# OmniLens++: Blind Lens Aberration Correction via Large LensLib Pre-Training and Latent PSF Representation

## Abstract

Emerging deep-learning-based lens library pre-training (LensLib-PT) pipeline offers a new avenue for blind lens aberration correction by training a universal neural network, demonstrating strong capability in handling diverse unknown optical degradations. This work proposes the OmniLens++ framework, which resolves two challenges that hinder the generalization ability of existing pipelines: the difficulty of scaling data and the absence of prior guidance characterizing optical degradation. To improve data scalability, we expand the design specifications to increase the degradation diversity of the lens source, and we sample a more uniform distribution by quantifying the spatial-variation patterns and severity of optical degradation. In terms of model design, to leverage the Point Spread Functions (PSFs), which intuitively describe optical degradation, as guidance in a blind paradigm, we propose the Latent PSF Representation (LPR). The VQVAE framework is introduced to learn latent features of LensLib's PSFs, which is assisted by a PSF-conditioned regularizer modeling the optical degradation process to constrain the learning of degradation priors. Experiments on diverse aberrations of real-world lenses and synthetic LensLib show that OmniLens++ exhibits state-of-the-art generalization capacity in blind aberration correction. Beyond performance, the AODLibpro is verified as a scalable foundation for more effective training across diverse aberrations, and LPR can further tap the potential of large-scale LensLib. The source code and datasets will be made publicly available.

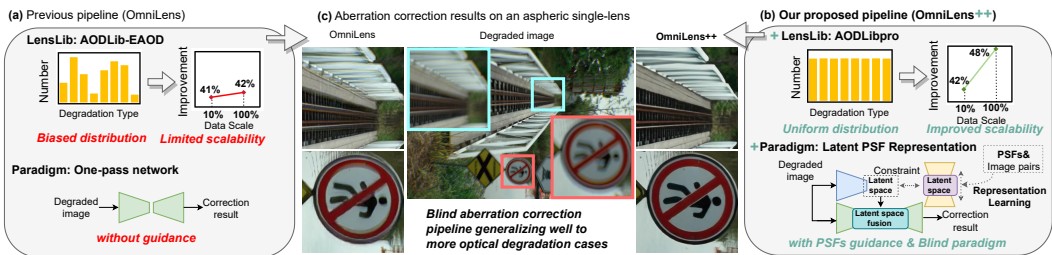

Figure 1: This work addresses the challenges of the current LensLib-PT pipeline. (a) OmniLens suffers from limited data scalability arising from the biased data distribution and lack of prior guidance for the model. (b) In OmniLens++, the proposed AODLibpro reveals uniform distribution contributing to improved scalability, while the latent PSF representation provides effective prior guidance with blind paradigm. (c) OmniLens++ effectively resolves the failure case of OmniLens.

## 1 Introduction

Lens aberrations, typically arising from compromised image quality optimization due to design trade-offs for specific requirements, *e.g.*, minimalist optical systems (Heide et al., 2013; Peng et al., 2019), or lenses on mobile devices (Chen et al., 2023), and manufacturing/assembly errors (Liu et al., 2024) in complex systems, introduce blur to the captured images. This blur is also referred to as optical degradation (Chen et al., 2021b), characterized by its distinctive spatially-varying nature where degradation varies across Field-of-Views (FoVs) and exhibits diverse patterns depending on

optical path, representing a fundamental image quality issue but has received limited attention in the learning and vision literature. With the advancement of image processing, computational post-processing (Schuler et al., 2011) has become a mainstream pipeline, also known as computational aberration correction. Unlike non-blind methods that rely on precise Point Spread Functions (PSFs) calibration (Chen et al., 2025), the blind pipeline (Schuler et al., 2012) offers more flexible and user-friendly advantages for users without optical expertise, where only the captured images are required for high-quality results. Recently, the deep learning-based Lens Library Pre-Training pipeline (LensLib-PT) has emerged as a powerful blind aberration correction paradigm (Gong et al., 2024). A universal network is trained to learn the mapping from diverse aberration distributions to clear images, demonstrating advantages over traditional blind deconvolution methods (Eboli et al., 2022) in terms of generalization to different optical degradation types. In particular, the OmniLens framework (Jiang et al., 2024b) delivers strong performance, enabled by a large-scale LensLib consisting of automatically designed lens samples to cover diverse real-world aberration distributions.

Building upon this paradigm, as shown in Figure 1, this paper further proposes *OmniLens++* as a robust solution to blind lens aberration correction, with particular focus on addressing two key challenges in OmniLens: 1) the *scalability* of LensLib is limited; 2) the aberration correction model lacks *guidance* to adaptively process diverse aberration distributions.

The OmniLens framework has laid the groundwork for high-quality LensLib construction by leveraging the Evolution-based Automatic Optical Design (EAOD) method (Jiang et al., 2024b) to generate large-scale lens sources. However, as shown at the top of Figure 1 (a), the constructed AODLib-EAOD reveals biased optical degradation distribution. This stems from several missing specifications in its lens-source generation process, and from which the applied sampling basis based on lens design indicators (average RMS spot radius) can hardly characterize the severity and spatial variation patterns of aberrations (Zhou et al., 2024). Consequently, the scalability of such a LensLib remains limited, where increasing the data scale fails to compensate for the absence of certain degradation types. To this end, we put forward a novel hybrid sampling basis to encompass any possible optical degradation patterns, which quantify the severity and spatial variation trends of optical degradation from the per-FoV image quality of lens imaging results rather than lens design indicators. As depicted at the top of Figure 1 (b), supplemented with enriched design specifications, the hybrid sampling contributes to the *large LensLib AODLibpro* with uniform aberration distributions without bias. The trained universal model can benefit from its scalability and achieve significant improvements with large-scale data. In addition, a synthetic benchmark is also established on the sampled lenses for evaluating aberration correction methods in terms of model design, which is also the first benchmark in this field for comprehensive evaluation across diverse aberration patterns.

Beyond constructing scalable data resources, an equally important challenge lies in designing models that can effectively handle diverse degradation types. Introducing degradation prior representation as guidance is a crucial design for such universal models (Potlapalli et al., 2023; Hu et al., 2025). In aberration correction, PSFs serve as an intuitive representation of degradation. Whether through explicit deconvolution (Lin et al., 2022) or implicit representation embedding (Jiang et al., 2024a), correction results can be effectively improved. However, these approaches require precise PSFs of the used lenses, which is infeasible in a blind pipeline. To this end, we propose the *Latent PSF Representation (LPR)* method, which predicts PSF information from degraded images in latent space, enabling *a blind paradigm while ensuring effective guidance of degradation priors* (at the bottom of Figure 1 (b)). LPR is motivated by encoding the optical priors embedded in PSFs, enabling their direct retrieval from degraded images to guide aberration correction. During representation learning, key PSF features modeling the optical degradation priors are learned and stored via self-supervised PSF reconstruction based on VQVAE (van den Oord et al., 2017) under the regularization of the forward operator of optical degradation. The Foundational Computational Aberration Correction (FoundCAC) model is then proposed, containing an independent encoder to extract degradation priors from images under the constraint of the learned LPR for latent space feature fusion.

Extensive experiments across diverse types of minimalist optical systems, misaligned lenses, high-end lenses, and our benchmark demonstrate that OmniLens++ achieves state-of-the-art performance in blind aberration correction. Taking an aspheric single-lens as an example, Figure 1 (c) illustrates that OmniLens++ can handle the failure case of OmniLens (Jiang et al., 2024b). Furthermore, we verify that: 1) scaling AODLibpro yields considerable improvements; 2) LPR-based guidance is more effective than other competing representation methods; and 3) LPR facilitates the model to sufficiently exploit the potential of large-scale AODLibpro.

The main contributions of this work are:

- We introduce OmniLens++, a LensLib-PT-based framework for blind aberration correction that achieves zero-shot generalization across diverse real-world lenses.
- We put forward AODLibpro, which enables a scalable data source by enriching the lens source coverage and sampling uniform-distributed optical degradation patterns.
- We propose the Latent PSF Representation (LPR) that enables the blind paradigm while injecting PSF-derived degradation priors to guide aberration correction.

## 2 RELATED WORK

**Lens aberration correction** through post-processing is widely applied in computational imaging, commonly used for Minimalist Optical System (MOS) imaging (Wei et al., 2024; Qian et al., 2025; Tseng et al., 2021) and image quality enhancement for mobile devices (Chen et al., 2023; Schuler et al., 2012). The non-blind methods with lens-specific paradigm (Chen et al., 2021b; Yanny et al., 2022) represent the current mainstream, but the repeated complex calibration (Chen et al., 2025) and model training (Chen et al., 2021a) for each different lens make it unfriendly to users without optical backgrounds. Blind pipelines offer a more flexible solution, requiring only degraded images as input. Traditional methods typically revolve around kernel estimation for non-blind deconvolution, compensated by natural image priors (Kee et al., 2011; Schuler et al., 2012; Yue et al., 2015; Eboli et al., 2022), but limited to handling mild aberrations, which can hardly generalize to more diverse lens types. The recent data-driven Lens Library Pre-Training (LensLib-PT) pipeline addresses this issue by training a universal model on a LensLib covering diverse aberrations. Early work (Li et al., 2021; Gong et al., 2024) leverages few manually collected lenses, where the scalability constraints result in limited coverage. While Zernike-based databases (Hu et al., 2021; Jiang et al., 2024c) contribute to data expansion, they suffer from shortcomings in the realism of distributions. In comparison, AODLib-EAOD constructed by EAOD algorithms in OmniLens achieves a balance between data scale and aberration distribution authenticity (Jiang et al., 2024b). Nevertheless, expanding the scale of AODLib-EAOD yields limited improvements, which is due to the data bias introduced by limited specifications and the simple sampling basis. To this end, this work incorporates more design specifications into AOD for broader coverage. A hybrid sampling basis is then designed based on the quantification of both severity and spatial-varying trends of optical degradation.

**Representation of degradation priors** is widely studied in All-in-One Image Restoration (AIO-IR) to guide models in processing different degradation types (Jiang et al., 2025a), yet remains under-explored in lens aberration correction. AIO-IR methods typically employ visual prompts (Potlapalli et al., 2023; Ma et al., 2023), contrastive learning (Li et al., 2022), text semantic prompts Ai et al. (2024), large model based feature extraction (Zhang et al., 2025), and pretext tasks (Hu et al., 2025) to characterize categorical information of different degradations. Unlike AIO-IR, different degradations in lens aberration correction all stem from convolution-induced image blur, varying primarily in severity and spatial-varying patterns, making category-based designs inapplicable. Given that PSFs directly characterize such degradations, several works propose using PSF information to guide aberration correction (Li et al., 2021; Lin et al., 2022; Jiang et al., 2024a; Luo et al., 2024). However, requiring precise PSFs during inference prevents these efforts from blind paradigms. To leverage PSF-based guidance while achieving blind operation, this work explores predicting PSF-related information from degraded images to guide the aberration correction. VQVAE (van den Oord et al., 2017) provides insights by utilizing VQ codebooks (Esser et al., 2021) in latent space to store key features, which has been investigated to represent optical degradation priors for guiding aberration correction Chen et al. (2023); Jiang et al. (2025b). Inspired by this, we propose LPR, which applies reconstruction to PSFs for storing key features, and contains a forward operator of optical degradation to constrain the codebook to learn latent PSF features describing optical priors.

## 3 METHODOLOGY

### 3.1 MOTIVATION

Building a Lenslib-PT-based blind aberration correction framework hinges on two axes: 1) the coverage and aberration distribution of Lenslib decides the application scope of the trained model; and

2) the model paradigm determines whether it can leverage the potential of the data. Therefore, we consider both aspects to train a foundational model with stronger generalization ability.

**LensLib construction.** Figure 1 (c) shows that the lack of the aspheric surface in specifications makes OmniLens (Jiang et al., 2024b) fail on such lenses, indicating that missing specifications lead to insufficient coverage of certain optical degradation distributions in the lens source. *More specifications should be included to increase the diversity of aberrations coverage during the lens source generation stage.* Meanwhile, Figure 1 (a) shows that expanding the scale of AODLib-EAOD from 10% to 100% yields limited improvements, highlighting the challenge of enlarging the data scale. This is mainly due to its RMS-based sampling basis, since RMS cannot describe the spatial-varying patterns of optical degradation, or even reflect its severity (Zhou et al., 2024). This leads to the fact that increasing the data scale fails to compensate for the absence of certain degradation types. *Designing a hybrid sampling basis that jointly considers spatial-varying properties and severity of optical degradation* comes to the forefront.

**Model paradigm.** Table 1 shows the performance of several existing aberration correction model paradigms, among which using Ground-Truth (GT) PSFs to guide the model demonstrates great potential. However, for the blind paradigm, its required PSF information is unavailable. Directly predicting PSFs from degraded images (the third row of Table 1) is an intuitive pipeline, but such a task is revealed to be an ill-posed problem (Joshi et al., 2008; Rego et al., 2021), which can hardly reach the per-

Table 1: Results of existing model paradigms on the benchmark set up in Section 3.2. Settings are detailed in Appendix C.

| Paradigm | Blind | PSNR | SSIM | LPIPS |
|---|---|---|---|---|
| Baseline | ✓ | 26.10 | 0.837 | 0.1701 |
| GT-PSFs-guided | ✗ | 28.73 | 0.873 | 0.1269 |
| PSFs prediction | ✓ | 28.25 | 0.861 | 0.1283 |
| PSFs feature prediction | ✓ | 28.42 | 0.867 | 0.1287 |

formance of GT-PSFs-guided pipeline. To this end, we attempt to predict PSF features in the latent space as guidance (the fourth row of Table 1). This strategy alleviates the prediction difficulty to some extent, but remains constrained by the large discrepancy between degraded images and PSFs. To further promote convincing PSF information prediction in latent space, *we aim to design an effective representation that retains informative PSF features* by leveraging the vector quantization and feature matching strategy in VQVAE (van den Oord et al., 2017).

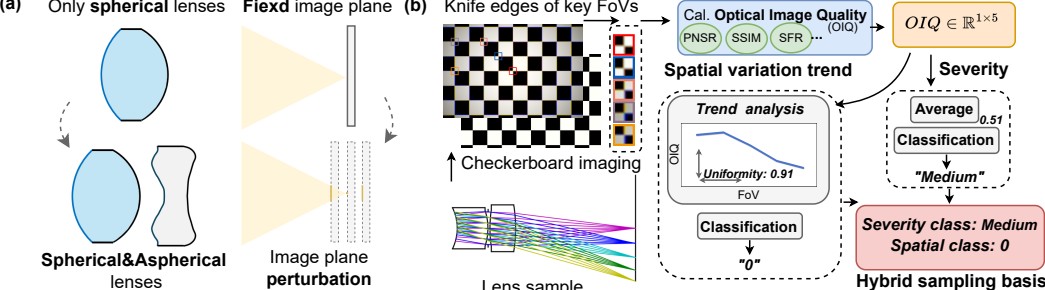

Figure 2: Illustration of the key designs in constructing AODLibpro. We expand surface type and imaging distance specifications in (a) to realize a broader set of optical degradation patterns during lens source generation; and quantify degradation severity and spatial variation trends via image quality assessment in (b), yielding a hybrid sampling that covers plausible optical degradation patterns.

### 3.2 DATA PREPARATION STAGE: LARGE LENSLIB CONSTRUCTION

**Generation of lens source.** To preserve diversity and realism in LensLib, we retain EAOD (Jiang et al., 2024b) for generating the lens source based on different sets of design specifications. We refer readers to (Jiang et al., 2024b) for details on EAOD. Building on this, we incorporate two influential yet previously omitted specifications to further diversify the lens source as shown in Figure 2 (a): (i) aspheric surface to broaden reachable degradation patterns, where high-order aspheric coefficients are added to lens parameters and EAOD's ray tracing is modified to handle aspheric surface case following (Chen et al., 2021b); and (ii) image distance, which shifts the focal field and shapes spatial degradation characteristics. To model the latter, we perturb the image distance of EAOD-optimized lenses with probability $\gamma$ within their depth of field to yield an additional variant sample. Finally, we feed all specifications into EAOD to search for candidate solutions to constitute the lens source. More details on the two specifications can be found in Appendix D.

**Hybrid sampling basis.** Since traditional design indicators do not fully capture the optical degradation of a lens in the image domain (§ 3.1), we quantify optical degradation by assessing the image quality of a lens's imaging results. We quantify per-FoV severity and categorize optical degradation by its variation trend, delivering a hybrid sampling basis to enable balanced sampling covering possible degradation patterns. Concretely, as in Figure 2 (b), a degraded checkerboard of the target lens and its paired GT are applied as the quantification base. Five knife-edge image patches sampled from the center to periphery FoVs are cropped for quantification. The Optical Image Quality (OIQ) is assessed to grade per-FoV degradation severity, where we adopt the weighted sum of fidelity metrics (PSNR and SSIM) and optical metric (SFR) as a prototype of OIQ. The average OIQ across the 5 FoVs is applied as the overall degradation severity of the target lens. For the entire lens source, we analyze the per-sample average OIQ distribution and partition the severity into 3 Severity-Class as *Strong*, *Medium*, and *Mild*. Furthermore, we categorize spatial variation patterns of optical degradation by defining 6 Spatial-Class from OIQ trends over FoVs. First, using the variance and mean of per-FoV OIQ, we compute the coefficient of variation to measure spatial uniformity $U_S$. Samples with values above a threshold $\alpha$ are labeled the *"spatial-uniform" Spatial-Class*. For $U_S < \alpha$, 5 *additional Spatial-Class* are defined by *the FoV of peak OIQ and the monotonicity of OIQ changes*. Detailed OIQ computation and Spatial-Class definitions are provided in Appendix D.

**Construction of AODLibpro.** With the novel sampling basis, we propose AODLibpro by sampling a balanced set from the lens source with uniform coverage over severity and spatial-variation patterns. We form 18 sub-classes by crossing the Severity-Class with Spatial-Class. From each sub-class, $m_1$ and $m_2$ instances are sampled to build AODLibpro `train` and `test`, respectively. The former offers large-scale training supervision, and the latter serves as a comprehensive benchmark to evaluate the performance of the potential networks across diverse aberration distributions.

### 3.3 STAGE I: LEARNING LATENT PSF REPRESENTATION

As shown in Figure 3, we pretrain a Latent PSF Representation (LPR) to encode rich optical priors from PSFs, where a VQVAE codebook materializes key features, and a PSF-conditioned regularizer supervises optical priors learning, supporting effective latent PSF feature prediction.

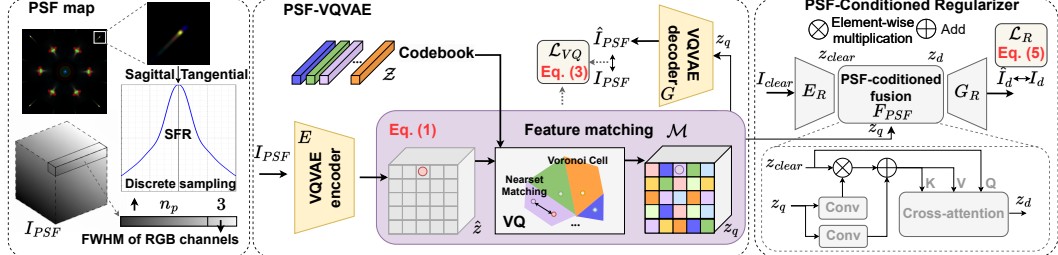

Figure 3: Stage I: Learning latent PSF representation. PSF-VQVAE explicitly stores the key latent PSF features constrained by the PSF-conditioned regularizer for modeling optical priors.

**PSF-VQVAE.** To represent PSFs as an information modality pixel-aligned with the degraded image $I_d \in \mathbb{R}^{H \times W \times 3}$, PSF kernels are converted into a PSF map $I_{PSF} \in \mathbb{R}^{H \times W \times N_p}$. For each pixel, we compress the PSF at its corresponding FoV and reshape it into a feature vector $x_p \in \mathbb{R}^{1 \times 1 \times N_p}$ along the channel dimension for insertion. To improve the information density of the compressed PSFs, following (Chen et al., 2023), we compute the SFR for PSFs supplemented with the PSFs' per-channel full width at half maximum (FWHM) to serve as $x_p$. The dimensionality of $x_p$ is $N_p = n_p + 3$, where $n_p$ is the resolution of the discrete sampling of the SFR curve. Then, the key lies in learning optical priors from the PSF map. VQVAE (van den Oord et al., 2017) learns a discrete latent prior into a codebook, providing transferable representations that can be directly retrieved. Motivated by this property, we propose PSF-VQVAE as the basis for LPR. The PSF map $I_{PSF}$ is mapped into the latent feature $\hat{z} \in \mathbb{R}^{H/8 \times W/8 \times n_z}$ with spatial size $(H/8 \times W/8)$ and channel dimension $n_z$ by the VQVAE encoder $E$, which is quantized by the codebook $\mathcal{Z} = \{z_k\}_{k=1}^K \in \mathbb{R}^{n_z}$ by finding the nearest neighbors in $\mathcal{Z}$ for its each element $\hat{z}_{ij}$, to calculate the discrete representation $z^q \in \mathbb{R}^{h \times w \times n_z}$:

$$z_{ij}^q = \arg \min_{z_k \in \mathcal{Z}} (\|\hat{z}_{ij} - z_k\|_2), \tag{1}$$

where $K$ denotes the size of the codebook and $i \in \{1, 2, \cdots, H/8\}$, $j \in \{1, 2, \cdots, W/8\}$ denote the coordinates in the feature space. The quantized PSF feature $z^q$ is then applied to reconstruct the

input PSF map $\hat{I}_{PSF}$ by the VQVAE decoder $G$ for self-supervised learning. The overall process is formulated as:

$$\hat{I}_{PSF} = G(z^q) = G(\mathcal{M}(E(I_{PSF}), \mathcal{Z})), \tag{2}$$

where $\mathcal{M}$ is the feature matching process. Following (van den Oord et al., 2017), reconstruction loss and codebook loss are applied as the objective function $\mathcal{L}_{VQ}$ for the training of PSF-VQVAE:

$$\mathcal{L}_{VQ}(E, G, \mathcal{Z}) = \|\hat{I}_{PSF} - I_{PSF}\|_1 + \|sg[\hat{z}] - z^q\|_2^2 + \beta\|sg[z^q] - \hat{z}\|_2^2, \tag{3}$$

where $sg[\cdot]$ denotes stop gradient operation that addresses the non-differentiability of $\mathcal{M}$ and facilitates the optimization of $E$ and $\mathcal{Z}$, and $\beta$ is set to 0.25 as a common practice.

**PSF-conditioned regularizer.** Given that the PSF-VQVAE learns raw PSF features without modeling optical priors, we introduce a PSF-conditioned regularizer to further facilitate LPR learning. The regularizer centers on injecting latent PSF features into the forward optical degradation process. We employ an end-to-end network to model this process, with the latent space conditioned on the quantized PSF features $z_q$ of the PSF-VQVAE to control the degradation pattern. An encoder $E_R$, sharing the same architecture as $E$, is applied to map the clear image $I_{clear}$ into latent space $z_{clear}$, which is fused with $z_q$ by the PSF-conditioned fusion module $F_{PSF}$ to produce the PSFs-conditioned optical degradation features $z_d$. To achieve global modulation in spatial domain of image features by PSF features, $F_{PSF}$ generates affine-based modulation and employs cross attention for feature conditioning. The degraded image $\hat{I}_d$ is then reconstructed from $z_d$ by a decoder $G_R$, where the objective function of the regularizer $\mathcal{L}_R$ is computed between $\hat{I}_d$ and the ground truth degraded image $I_d$. The regularizer and its training objective can be formulated as:

$$\hat{I}_d = Regularizer(I_{clear}; z^q) = G_R(z_d) = G_R(F_{PSF}(E_R(I_{clear}), z^q)), \tag{4}$$

$$\mathcal{L}_R(E, \mathcal{Z}, E_R, G_R, F_{PSF}) = \|\hat{I}_d - I_d\|_1. \tag{5}$$

To this end, the LPR is achieved by jointly training the PSF-VQVAE and the PSF-conditioned regularizer in an end-to-end manner, with the overall training objective given by:

$$\mathcal{L}_{LPR} = L_{VQ} + L_R. \tag{6}$$

Within LPR, optical priors are implicitly captured by the codebook and the regularizer, which are subsequently leveraged to guide the prediction of latent PSF features from degraded images.

### 3.4 STAGE II: TRAINING A FOUNDATIONAL MODEL

Figure 4 shows the established Foundational Computational Aberration Correction (FoundCAC) model guided by the pre-trained LPR. Considering that we aim to predict PSF features in latent space to guide aberration correction, FoundCAC mainly adopts an UNet architecture $\{E_C, G_C\}$ identical to that of VQVAE as the baseline model.

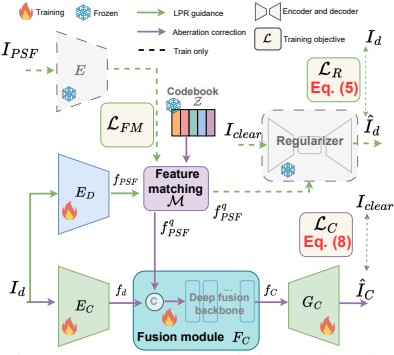

Figure 4: Stage II: Training a foundational model.

**Latent PSF features prediction.** To predict latent PSF features characterizing optical priors from degraded inputs $I_d$, we employ an encoder $E_D$ identical to $E_C$ constrained by the learned LPR. The extracted features $f_{PSF}$ are then matched with the pre-trained LPR codebook $\mathcal{Z}$ to retrieve the corresponding PSF features $f_{PSF}^q$ of $I_d$. During training, the GT PSF map $I_{PSF}$ is passed through the trained PSF-VQVAE to produce the GT LPR $z_{gt} = \mathcal{M}(E(I_{PSF}), \mathcal{Z})$ for supervising the predicted $f_{PSF}^q$ based on feature matching loss $\mathcal{L}_{FM}$ following (Chen et al., 2022a). Additionally, the trained regularizer is applied to further supervise the prediction of $f_{PSF}^q$. Conditioning the regularizer on $f_{PSF}^q$, we degrade $I_{clear}$ to $\hat{I}_d$ and compute $\mathcal{L}_R$ of Equation 5 against $I_d$. In this way, the training objective of LPR-guided latent PSF features prediction $\mathcal{L}_{PFP}$ is the combination of codebook-guided $\mathcal{L}_{FM}$ and regularizer-guided $\mathcal{L}_R$:

$$\mathcal{L}_{PFP} = \mathcal{L}_{FM} + \mathcal{L}_R. \tag{7}$$

**PSF-features-conditioned network.** The latent space of FoundCAC is conditioned on the predicted PSF features for leveraging the learned optical priors. We introduce a fusion module $F_C$ that uses $f_{PSF}^q$ to modulate the features $f_d$ extracted by $E_C$. We concatenate $f_{PSF}^q$ with $f_d$, and feed the results into several backbone blocks to perform deep fusion, yielding the computational aberration correction features $f_C$. The backbone can be initialized with any backbone in low-level vision, such as the classical RRDB (Wang et al., 2018) and RSTB (Liang et al., 2021) blocks. Finally, $G_C$ decodes $f_C$ to reconstruct the correction result $\hat{I}_C$. Similar to Chen et al. (2021a;b); Gong et al. (2024), we supervise $\hat{I}_C$ using an objective $\mathcal{L}_C$ that combines fidelity and perceptual losses:

$$\mathcal{L}_C = \|\hat{I}_C - I_{clear}\|_1 + \|\phi(\hat{I}_C) - \phi(I_{clear})\|_2^2, \tag{8}$$

where $\phi$ is a pre-trained VGG-16 network (Simonyan & Zisserman, 2015). We do not discuss GAN or diffusion generative paradigms here, since the aberration correction task commonly focuses more on the modeling and learning of degradation patterns.

The entire FoundCAC model is trained end-to-end under the joint supervision of $\mathcal{L}_C$ and $\mathcal{L}_{PFP}$. During this stage, only $E_C$, $G_C$, $F_C$, and $E_D$ are updated, while the parameters of the pre-trained codebook and the regularizer in LPR remain frozen.

## 4 EXPERIMENTS

### 4.1 IMPLEMENTATION DETAILS

**Datasets.** We sample $m_1 = 200$ and $m_2 = 3$ instances per class for AODLibpro `train` and AODLibpro `test`, yielding $3,600$ training lenses and $54$ test lenses with no overlap. For AODLibpro `train`, each lens randomly degrades $40$ images from Flickr2K (Timofte et al., 2017) via a precise imaging simulator (Yang et al., 2024). For AODLibpro `test`, the degraded images are synthesized based on additional 26 collected clear images. Meanwhile, through manually designing and gathering open source designs, we construct a real-world lens aberration dataset *RealLens-Sim*, consisting of 4 minimalist optical systems with spherical or aspheric surfaces (MOS-A/S) (Jiang et al., 2024b), 1 metalens from Tseng et al. (2021) (MOS-Meta), one smartphone lens under 3 misalignment (MA) cases, and 2 high-end lenses including an ultra-wide lens and a smartphone lens (High-end). The same simulator is applied to simulate the paired images of them for numerical evaluation. For real-world cases, we collect real-snapped optical degradation images *RealLens-Snap* with fabricated single lenses, the commercial minimalist lens $CAYE\ 50mm\ f1/4$, two DSLRs of $Canon\ 24mm\ f1/4$ and $Sony\ 18135$, and nano-optics data from Tseng et al. (2021).

**Training details.** On AODLibpro `train`, we first pre-train LPR for $100K$ iterations, and train the FoundCAC model for $200K$ iterations. Both training stages use $256 \times 256$ random crops with flips and rotations for augmentation, with a batch size of 16. We use Adam with a learning rate of $2e-4$ to $1e-6$ decayed by cosine annealing. Training is conducted on two RTX 4090 GPUs, and inference uses a single RTX 4090. Consistent with Chen et al. (2022a), all encoders and decoders employ 3 groups of ResBlocks. For FoundCAC, we use 4 RSTB layers from Swin-T (Liang et al., 2021) as the deep fusion backbone, given its strong performance on spatial-varying degradation (Jiang et al., 2024a). Ablations on alternative backbones are reported in Section 4.2 and 4.4.

In the following sections, we apply *RealLens* to evaluate the performance of blind aberration correction pipelines, and we further benchmark the model paradigms via AODLibpro `Test` with emphasis on the PSF representations. For more implementations, please refer to Appendix E and F.

### 4.2 BLIND CORRECTION FOR REAL-WORLD LENS ABERRATIONS

**Numerical evaluation on *RealLens-Sim*.** On the *RealLens-Sim*, we evaluate the overall capability of blind lens aberration correction methods to handle real-world aberrations as shown in Table 2. The suite includes the state-of-the-art deconvolution method fast two-step (Eboli et al., 2022), a universal Image Restoration (IR) model for real-world degradations represented by S3Diff (Zhang et al., 2024) trained under the BSRGAN (Zhang et al., 2021) data regime, and various LensLib-PT methods containing LensLibs of ZEBASELib (Gong et al., 2024), ZernikeLib (Jiang et al., 2024c), AODLib-LensNet (Côté et al., 2021), and AODLib-EAOD in OmniLens (Jiang et al., 2024b). For

the LensLib-PT methods, we adopt SwinIR as the network architecture for its superior overall performance in OmniLens. In addition to our final FoundCAC trained on AODLibpro, we also report SwinIR trained on AODLibpro as a reference result.

Overall, our full framework achieves state-of-the-art blind aberration correction results, generalizing well across diverse real-world lens aberrations. OmniLens++ delivers pronounced gains on challenging aberration cases such as MOS-S/A and MOS-Meta, robustly handles highly stochastic misalignment aberrations, and surpasses fast two-step methods on high-end lens aberrations, which are specifically designed for them. Notably, a universal IR model trained without optical degradation data copes with the relatively mild aberrations of high-end lenses, yet struggles on more complex and severe cases, indicating the importance of dedicated research on lens aberration correction. Finally, the last three rows show that, atop the OmniLens baseline, AODLibpro and LPR yield average PSNR improvements of $0.71dB$ and $0.81dB$ respectively, verifying that both our data and model paradigm designs enhance the capacity of LensLib-PT-based pipeline for blind aberration correction. Additional visual results are provided in Appendix G.1.

Table 2: Comparison with potential blind lens aberration correction pipelines on *RealLens-Sim*. We report the PSNR/SSIM/LPIPS results under each sub-test-lenses-set. The latency of each method to process an image of $1920 \times 1280$ is also provided. The **best** and second results are highlighted.

| Method | Latency (s) | *RealLens-Sim* | | | | |
| --- | --- | --- | --- | --- | --- | --- |
| | | MOS-S/A | MOS-Meta | MA | High-end | Average |
| Fast two-step | **0.390** | 20.65/0.712/0.3210 | 21.56/0.648/0.4596 | 27.14/0.765/0.1861 | 28.49/0.835/0.1650 | 24.46/0.740/0.2829 |
| Universal IR model (S3Diff) | 9.933 | 20.34/0.746/0.2615 | 21.36/0.687/0.4229 | 26.90/0.799/0.1779 | **29.69**/0.857/0.1423 | 24.57/0.772/0.2512 |
| ZEBASELib-PT | 0.782 | 23.09/0.791/0.2969 | 18.14/0.679/0.4931 | 25.91/0.837/0.1274 | 28.16/0.901/0.1053 | 23.83/0.802/0.2557 |
| ZernikeLib-PT | 0.782 | 24.41/0.822/0.1841 | 20.50/0.707/0.3806 | 25.16/0.850/0.1248 | 27.23/0.898/**0.0932** | 24.33/0.819/0.1957 |
| AODLib-LensNet-PT | 0.782 | 23.10/0.796/0.2937 | 18.56/0.678/0.4993 | 25.97/0.858/0.1224 | 27.49/**0.904**/0.0977 | 23.78/0.809/0.2533 |
| AODLib-EAOD-PT | 0.782 | 26.72/0.853/0.1597 | 21.96/0.748/0.3671 | 26.02/0.861/0.1015 | 27.72/0.903/0.0953 | 25.14/0.839/0.1842 |
| AODLibpro-PT (SwinIR) | 0.782 | **27.11**/0.862/0.1542 | 21.80/0.752/0.3441 | 26.69/**0.866**/0.1035 | 27.81/**0.904**/0.1006 | 25.85/0.846/0.1756 |
| **AODLibpro-PT (FoundCAC)** | 0.417 | 27.09/**0.864/0.1465** | **23.32/0.769/0.3145** | **27.44/0.866/0.0982** | 28.81/**0.904**/0.0975 | **26.66/0.851/0.1642** |

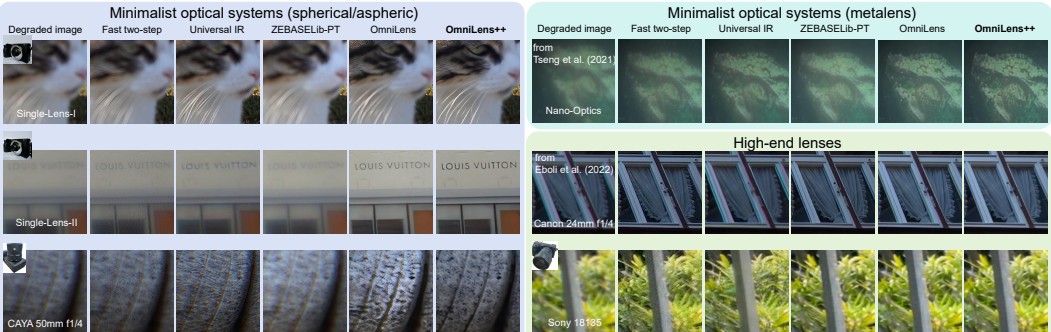

Figure 5: Visual results of representative blind lens aberration correction pipelines across real-world cases. More results and the capture details are provided in Appendix G.5 and G.6.

**Qualitative evaluation on *RealLens-Snap*.** Using real-snapped images (*RealLens-Snap*), we further conduct qualitative validation of the representative methods in Table 2, as shown in Figure 5. OmniLens++ shows clear advantages in minimalist lenses, reflected in better handling of blur, stronger suppression of purple fringing, and little introduction of artifacts and ringing. For out-of-domain metalens imaging results, our method successfully recovers a certain amount of high-frequency detail and removes most chromatic aberration. Finally, similar to the fast two-step method, OmniLens++ also improves the image quality of high-end DSLR lenses by enhancing sharpness and correcting purple fringing. In summary, OmniLens++ exhibits strong generalization capacity for blind aberration correction on real-snapped images.

## 4.3 EFFECTIVENESS OF FOUNDATIONAL ABERRATION CORRECTION MODEL AND LPR

**Comparison with state-of-the-art aberration correction networks.** We fix the training set to AODLibpro train and evaluate on the AODLibpro test benchmark, comparing FoundCAC against state-of-the-art blind aberration correction networks (Wang et al., 2018; Liang et al., 2021; Cho et al., 2021; Chen et al., 2022b; Zamir et al., 2022; Chen et al., 2024; Potlapalli et al., 2023;

Table 3: Comparison with state-of-the-art networks in CAC on AODLibpro `test` benchmark. The **best** and second results are highlighted.

| Network | Latency (s) | PSNR | SSIM | LPIPS |
|---|---|---|---|---|
| RRDBNet | 0.296 | 27.22 | 0.852 | 0.1519 |
| SwinIR | 0.782 | 28.29 | 0.871 | 0.1346 |
| MIMOUnet | 0.405 | 28.11 | **0.874** | 0.1651 |
| NAFNet | 0.353 | 27.48 | 0.866 | 0.2078 |
| Restormer | 1.859 | 27.11 | 0.867 | 0.1430 |
| X-Restormer | 2.797 | 28.11 | 0.870 | 0.1408 |
| PromptIR | 2.063 | 27.14 | 0.869 | 0.1404 |
| DiffBIR | 66.130 | 27.52 | 0.833 | 0.1430 |
| S3Diff | 9.933 | 23.10 | 0.762 | 0.1678 |
| FOVKPN | 0.166 | 27.23 | 0.851 | 0.1586 |
| DFUnet | **0.137** | 27.04 | 0.841 | 0.1639 |
| **FoundCAC (RRDB)** | 0.330 | 26.63 | 0.858 | 0.1409 |
| **FoundCAC (Swin)** | 0.417 | **28.67** | 0.873 | **0.1277** |

Table 4: Comparison between LPR and other potential representations for PSFs. We employ the same fusion module as LPR for all the PSF features (the same is true for the experiments in Table 1).

| Representation | PSNR | SSIM | LPIPS |
|---|---|---|---|
| Baseline | 26.10 | 0.837 | 0.1701 |
| GT-PSFs-guided (SFR) | 28.73 | 0.873 | 0.1269 |
| GT-PSFs-guided (Downsample) | 28.20 | 0.861 | 0.1315 |
| PSFs prediction | 28.25 | 0.861 | 0.1283 |
| PSFs feature prediction | 28.42 | 0.867 | 0.1287 |
| Regularizer (Degradation) | 28.40 | 0.864 | **0.1273** |
| PSF-VQVAE | 28.55 | 0.868 | 0.1293 |
| +Regularizer (Correction) | 26.95 | 0.862 | 0.1346 |
| **+Regularizer (Degradation)** | **28.67** | **0.873** | 0.1277 |
| +Regularizer (Both applied) | 26.76 | 0.861 | 0.1395 |

Lin et al., 2024; Zhang et al., 2024; Chen et al., 2021a;b) in Table 3. We also report a CNN version of FoundCAC that adopts RRDB as the deep fusion backbone. Compared with their baselines, FoundCAC using RRDB and RSTB as backbones shows better correction performance without a noticeable increase in inference time. This indicates the effectiveness of our LPR-guided model paradigm and its applicability across multiple backbone architectures. Meanwhile, with RSTB as the backbone, which is suited for spatial-varying optical degradation, FoundCAC achieves state-of-the-art results on the benchmark. PromptIR shows limited improvement over its baseline Restormer, suggesting that the class-based prompt is not appropriate for representing optical priors. Finally, diffusion-based methods perform poorly on diverse anerrations with a clear disadvantage in latency. This verifies that for aberration correction, learning optical priors is more effective than injecting clear image priors. Furthermore, the representation of optical priors enables FoundCAC to outperform methods with specialized network designs for optical degradation (FOVKPN and DFUnet).

**Comparison with potential PSF representations.** In Table 4, we further explore how to learn a better latent PSF representation. First, preparing PSF maps in the form of SFR outperforms downsampling in Jiang et al. (2024a). With this setup, constraining with PSF-VQVAE or with PSF-conditioned regularizer individually yields better guidance for latent PSF features prediction than direct supervision on PSFs or their features, validating both designs. Their combination achieves the best performance by leveraging the ability of the VQ strategy to store feature vectors and the advantage of the regularizer in modeling optical priors. In addition, another potential solution that constrains the latent space of the PSF-VQVAE directly with the aberration correction task is compar, failing to induce effective PSF representations. Overall, the proposed LPR provides more effective guidance to aberration correction than alternative representations.

## 4.4 ABLATION STUDIES

We perform ablations to investigate the individual effectiveness of AODLibpro and LPR, and their interaction within OmniLens++. To ensure fairness, any experiment that modifies the training data is tested on *RealLens-Sim* to assess the overall framework, while models trained on the full AODLibpro `train` are evaluated on the AODLibpro `test` to evaluate the model paradigm.

Table 5: Ablations on design specifications. A.S.: Aspheric Surface. I.P.P: Image Plane Perturbation.

| Specification | PSNR | LPIPS |
|---|---|---|
| Baseline in OmniLens | 25.95 | 0.1897 |
| + A.S. | 25.72 | 0.1893 |
| **+ A.S.&I.P.P.** | 25.85 | 0.1756 |

Table 6: Ablations on sampling basis.

| Sampling basis | PSNR | LPIPS |
|---|---|---|
| RMS | 25.56 | 0.1843 |
| Severity-Class | 26.10 | 0.1859 |
| Spatial-Class | 25.85 | 0.1815 |
| **Hybrid** | 25.85 | 0.1756 |

Table 7: Ablations on components of LPR guidance.

| | $E_D$ | Fusion | PSNR | LPIPS |
|---|---|---|---|---|
| Baseline | ✗ | ✗ | 26.10 | 0.1701 |
| 1 | ✓ | ✗ | 25.43 | 0.1675 |
| 2 | ✗ | RRDB | 26.42 | 0.1466 |
| 3 | ✓ | RRDB | 26.63 | 0.1409 |
| 4 | ✗ | Swin | 28.46 | 0.1318 |
| 5 | ✓ | Swin | 28.67 | 0.1277 |

**Evaluation of AODLibpro.** We train a SwinIR model under different LensLib settings to verify their performance. Table 5 shows that incorporating additional design specifications can improve the quality of LensLib. The results in Table 6 indicate that, compared with the RMS-based sampling basis, the proposed Severity-Class and Spatial-Class yield models of stronger generalization ability, and combining them contributes to a more comprehensive improvement. We further demonstrate the advantages of AODLibpro over its baseline AODLib-EAOD in Figure 6. Benefiting from specification expansion and the hybrid sampling basis, the samples in AODLibpro are uniformly distributed

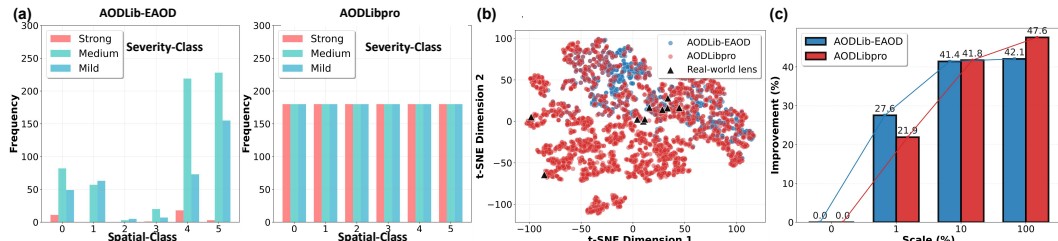

Figure 6: AODLibpro *v.s.* AODLib-EAOD in terms of uniformity of aberration distributions, coverage over real-world lenses, and scalability. (a) Histogram of degradation type distributions in the sampled lenses. (b) LensLib coverage visualization based on OIQ evaluated per FoV and wavelength. (c) Improvements of SwinIR trained with LensLibs of different scales over the method without LensLib (Eboli et al., 2022). The improvement is averaged across the PSNR and LPIPS.

across degradation severity and spatial variation patterns (Figure 6 (a)), while the overall aberration distribution is broader and can cover all of the lens samples in *RealLens-Sim* (Figure 6 (b)). These advantages are conducive to the scalability of AODLibpro, where significant gains are brought by increasing the data scale (Figure 6 (c)), addressing the limitations of AODLib-EAOD.

**Evaluation of components for LPR guidance.** As shown in Table 7, starting from the UNet baseline, we evaluate the effectiveness of the predicted latent PSF features and the proposed fusion module. Settings 1, 3, and 5 show that both CNN-based and Transformer-based fusion enable LPR to provide effective guidance, yielding PSNR gains of $0.32 \sim 2.57dB$. Meanwhile, the deep fusion backbone is necessary because simply adding the predicted latent PSF features yields limited improvements, indicating that the model requires considerable parameters to learn the interaction between optical priors and image features. The comparisons between settings 2 and 3, and between 4 and 5, further show that the effective guidance of LPR is not due to the additional parameters; under the same backbone, incorporating the latent PSF features contributes to significant gains.

Table 8: Improvements brought by LPR under different data scales of AODLibpro.

| LPR | Sclae: 1% | | Sclae: 10% | | Sclae: 100% | |
|---|---|---|---|---|---|---|
| | PSNR | LPIPS | PSNR | LPIPS | PSNR | LPIPS |
| ✗ | 23.82 | 0.2187 | 25.23 | 0.1998 | 25.92 | 0.1881 |
| ✓ | 23.39 (↓10.41%) | 0.2363 (↓8.05%) | 25.72 (↑10.67%) | 0.1782 (↑10.81%) | 26.66 (↑**15.67%**) | 0.1642 (↑**12.71%**) |

**Effectiveness of LPR under different data scales.** We investigate whether the proposed LPR-guided model paradigm can exploit the potential of the enlarged AODLibpro in Table 8. With 1% data, LPR fails to learn effective PSF representations and produces negative gains. This is anticipated, as the codebook captures only a limited set of PSF features, which prevents reliable retrieval for unseen real-world distributions. Scaling the data alleviates this and enables LPR to provide effective guidance. As data scale increases, LPR delivers larger improvements over the baseline, indicating that it effectively leverages the enhanced scalability of AODLibpro.

## 5 CONCLUSION AND FUTURE WORK

**Conclusion.** We present OmniLens++ as a robust blind aberration correction solution under the LensLib-PT paradigm. AODLibpro improves coverage and scalability on the data side, and LPR provides the first PSF-guided model paradigm that preserves the blind setting. OmniLens++ generalizes across diverse real-world aberrations, which is comparable to the state-of-the-art deconvolution-based method on mild high-end lens aberrations, and excels on severe aberrations in minimalist optical systems, positioning it as a promising general solution for blind aberration correction.

**Future work.** Looking ahead, benefiting from the scalability of AODLibpro, future work will focus on extending it with metasurfaces and diffractive optical elements for stronger generalization. Then, building on the OmniLens++ framework, we will further investigate aberration correction under degradations coupled with optics including depth of field (Abuolaim & Brown, 2020; Ruan et al., 2022; Yang et al., 2023; 2025), under display imaging (Feng et al., 2021; Wang et al., 2024), low light conditions (Liu et al., 2023), and sensor noise (Zheng et al., 2025) to support real-world deployment. Last but not least, we believe it is important to exploit FoundCAC as a pretrained model, where a flexible fine-tuning pipeline is urged to enhance its practical value.

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

# A STATEMENT

**Ethics statement.** We explore a robust blind aberration correction framework to address lens aberrations of arbitrary lenses, which is valuable for the applications of light-weight optical systems with uncorrected aberrations. This technology will facilitate the development of various mobile and wearable devices, such as mobile cameras, intelligent robots, and AR/VR equipment. However, the advantage of the tiny size of light-weight optical systems may also be leveraged for military reconnaissance and sneak shots. We hope that the relevant applications can be regularized by laws to prevent misuse.

**Reproducibility statement.** We state that the OmniLens++ framework is highly reproducible. At the data level, the specific specification settings, sampling basis, and optical degradation image preparation details are presented in Section 3.2 and Appendix D, F.1, and the simulation component directly uses the open source DeepLens (Yang et al., 2024) repository. At the model level, Section 3.3 describes feature processing at each key node in detail, and the exact forms of all loss functions are provided. Appendix E also presents the specific network architectures and parameter settings for each stage of FoundCAC. Upon publication, all source code and data (lens design files, simulated and real-snapped optical degradation images) will be released. Together with the detailed descriptions in the paper, the entire OmniLens++ framework can be well reproduced.

**Large language model statement.** We employed a large language model tool for language editing only, such as improving fluency, tightening long sentences, and refining phrasing to achieve a more native-like expression. All technical claims, methods, and results were written, verified, and approved by the authors, who assume full responsibility for the final manuscript.

# B ILLUSTRATION OF ABBREVIATIONS

Due to some lengthy proper terms, this paper uses and defines many abbreviations. Table 9 summarizes all abbreviations along with their full forms to facilitate quick reference.

Table 9: Illustration of abbreviations.

| Abbreviation | Explanation |
| --- | --- |
| FoV | Field of View |
| PSF | Point Spread Function |
| LensLib-PT | Lens Library Pre-Training |
| FoundCAC | Foundational Computational Aberration Correction Model |
| AOD | Automatic Optical Design |
| EAOD | Evolution-based Automatic Optical Design |
| OIQ | Optical Image Quality |
| LPR | Latent PSF Representation |
| GT | Ground Truth |
| IR | Image Restoration |
| MOS | Minimalist Optical System |
| RSTB | Residual Swin Transformer Block |
| RRDB | Residual in Residual Dense Block |

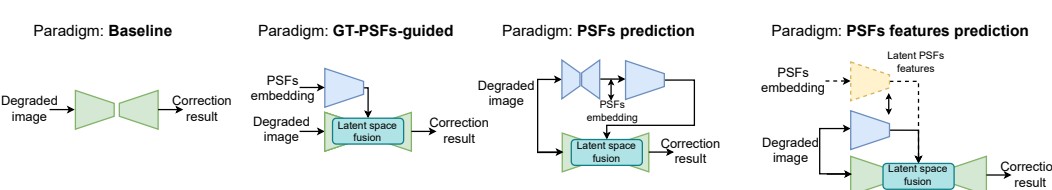

Figure 7: Illustration of the model paradigms in motivation.

# C DETAILED SETTINGS FOR EXPERIMENTS IN MOTIVATION

The schematic diagrams of the model paradigms in Table 1 of Section 3.1 are shown in Figure 7. All encoders and decoders use the same design as in the final FoundCAC, namely 3 groups of ResBlocks. All fusions between PSF features and image features adopt $F_C$, that is, concatenation

followed by deep feature backbones of $4$ RSTB layers. Supervision for both PSFs prediction and PSFs features prediction applies L1 loss. Other training settings follow those of the aberration correction stage in FoundCAC. Additionally, all models in this table are trained on AODLibpro-Train and evaluated on the AODLibpro-Test benchmark to evaluate network performance.

# D    DETAILS FOR AODLIBPRO CONSTRUCTION

## D.1    DETAILS FOR SUPPLEMENTED SPECIFICATIONS

**Definition of the aspheric surface.** The aspheric lens surface is an optical surface whose curvature deviates from a constant-radius sphere. Unlike conventional spherical surfaces, its profile is mathematically defined by higher-order polynomials, enabling precise control over light refraction across the entire aperture. The height of a standard aspheric surface Sun et al. (2021) is defined as a function of the radial distance $r$:

$$h(r) = \frac{cr^2}{1 + \sqrt{1 - (1 + \kappa)c^2 r^2}} + \sum_{i=2}^{N_A} a_{2i} r^{2i}, \tag{9}$$

where $c$ denotes the curvature, $\kappa$ is the conic coefficient, $a_{2i}$'s are higher-order coefficients, and $N_A$ defines the highest-order aspheric coefficient.

**Image distance perturbation constrained by depth of field.** We constrain the image distance perturbation amplitude within acceptable limits using the Depth of Field (DoF) formula:

$$\Delta L = \Delta L_1 + \Delta L_2 = \frac{F\delta L^2}{f^2 + F\delta L} + \frac{F\delta L^2}{f^2 - F\delta L}, \tag{10}$$

where $\delta$ is the permissible circle of confusion diameter, $f$ is the lens focal length, $F$ is the F-number, $L$ is the image distance, $\Delta L_1$ is the near DOF, and $\Delta L_2$ is the far DOF. Image distance perturbation range is constrained within $[-\Delta L_1, \Delta L_2]$, and $\delta$ is set to $24\mu m$ in this work. The perturbation probability $\gamma$ is set to $25\%$ empirically.

## D.2    DETAILS FOR HYBRID SAMPLING BASIS

**Calculation of OIQ.** OIQ incorporates traditional fidelity-based image quality metrics (PSNR and SSIM (Wang et al., 2004)) as well as the SFR-based metric to provide an image quality assessment considering optical properties:

$$OIQ = \lambda_1 \frac{PSNR}{50} + \lambda_2 \frac{SSIM - 0.5}{0.5} + \lambda_3 \, OIQE, \tag{11}$$

where PSNR and SSIM are processed following (Liang et al., 2024) to obtain normalized metrics, while SFR is represented as OIQE (Jiang et al., 2024a), which denotes the ratio of the SFR of the evaluated target to that of a lens without optical degradation. The weights are set to $\lambda_1 = 0.4, \lambda_2 = 0.3, \lambda_3 = 0.3$ to control the value ranges so that the $3$ normalized metrics are close in range following (Liang et al., 2024).

**Definition of Severity-Class.** The average OIQ across the $5$ knife-edge image patches of different FoVs is calculated to represent the overall severity level of the target lens's optical degradation. The average OIQ lies within $[0, 1]$, where a larger value indicates lower overall optical degradation severity (higher optical image quality). Therefore, we divide the range into $3$ intervals to categorize different optical degradation severity levels for sampling, as shown in Figure 8.

**Calculation of the spatial uniformity.** We first define the coefficient of variation (CV) from the variance and mean of OIQ across the $5$ FoVs, then use it to compute a spatial uniformity metric $U_S$ that measures the uniformity of the optical degradation spatial distribution:

$$CV = \frac{Std(\{OIQ_i | i = 1, ..., 5\})}{Avg(\{OIQ_i | i = 1, ..., 5\})}, \tag{12}$$

$$U_S = e^{-5CV}, \tag{13}$$

where $OIQ_i$ is the OIQ of the $i_{th}$ FoV. A larger $U_S$ indicates that the optical degradation spatial distribution of the lens is more uniform.

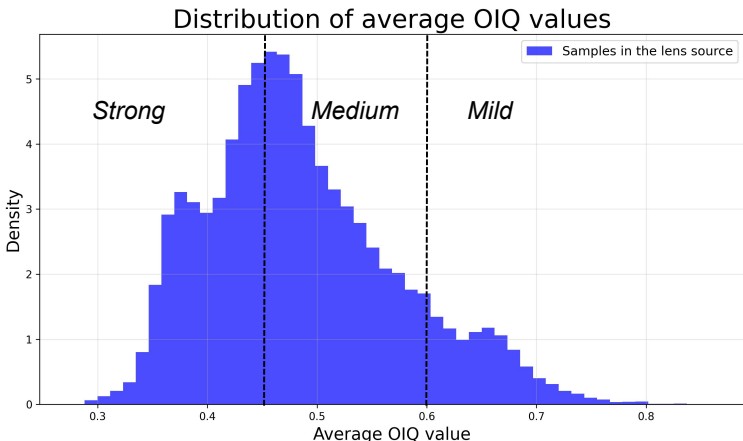

Figure 8: Illustration of the definition for Severity-Class.

| Spatial-Class | $U_S$ | OIQ trend | Classification criteria | Descriptions |
|---|---|---|---|---|
| 0 | $U_S > \alpha$ | | $U_S > \alpha$ | Spatial-uniform |
| 1 | $U_S < \alpha$ | | $U_S < \alpha, all\ Diff(\textbf{OIQ}) < 0$ | OIQ monotonically decreases, and degradation becomes progressively more severe from the central FoV toward the periphery. |
| 2 | $U_S > \alpha$ | | $U_S < \alpha, min\_idx(\textbf{OIQ}) = 1$ | Degradation is most severe at the central FoV, that is, the minimum OIQ occurs at the "1" FoV (including the case where OIQ increases monotonically). |
| 3 | $U_S > \alpha$ | | $U_S < \alpha, min\_idx(\textbf{OIQ}) = 2/3/4$ | Degradation is most severe in the middle three FoV, that is, the minimum OIQ occurs at FoV "1", "2", or "3". |
| 4 | $U_S > \alpha$ | | $U_S < \alpha, min\_idx(\textbf{OIQ}) = 5,$ $max\_idx(\textbf{OIQ}) = 1,$ | Degradation is most severe at the peripheral FoV and mildest at the center (unlike the monotonically decreasing case, the OIQ across the three middle FoVs is non-monotonic). |
| 5 | $U_S > \alpha$ | | $U_S < \alpha, min\_idx(\textbf{OIQ}) = 5,$ $max\_idx(\textbf{OIQ}) \neq 1,$ | Degradation is most severe at the peripheral FoV and not mildest at the center. |

We set $\alpha = 0.85$ emperically.
$all\ Diff(\textbf{OIQ}) < 0$: The first order difference of the OIQ vector is strictly positive.
$min\_idx(\textbf{OIQ})/max\_idx(\textbf{OIQ})$: The FoV index at which the minimum/maximum OIQ occurs.

Figure 9: Illustration of the definition of Spatial-Class. Notably, for cases where degradation is most severe at the peripheral FoV, we define Spatial-Classes "4" and "5" because we observe that some lenses, despite showing the most severe degradation at the peripheral FoV, also exhibit relatively severe degradation at the center FoV. This degradation pattern clearly differs from the typical degradation pattern of a sharp center and blurred edge, which should be independently considered.

**Definition of Spatial-Class.** Figure 9 provides detailed definitions for each Spatial-Class, including the $U_S$ value range, schematic OIQ trend plots, classification criteria, and supplementary descriptions. This basis allows any OIQ to fall into one class, enabling the classification and description of all possible optical degradation patterns. To the best of our knowledge, there is currently no such detailed categorization of optical degradation distribution patterns, so we make a preliminary attempt to explore this problem here. Table 6 shows that using the proposed Spatial-Class for sampling can construct a more effective LensLib. We hope this classification approach can provide new insights for this field to understand optical degradation distribution patterns.

**Discussion on chromatic aberrations.** We do not include chromatic aberration as a criterion in optical degradation classification because preliminary experiments show that it has little impact on

the final correction results, as shown in Figure 10. Using the same computation pipeline as $U_S$, we compute the channel-wise uniformity of OIQ to quantify the severity of chromatic aberration and divide it into 5 categories, where a smaller category index indicates more severe chromatic aberration. The distribution of chromatic aberration in AODLibpro `Test` is shown in Figure 10 (a). We tally the performance of our trained FoundCAC at different chromatic levels and find no obvious pattern in its performance as the severity of chromatic aberration changes, as shown in Figure 10 (b). This indicates that chromatic aberration has little effect on aberration correction models trained on LensLib, even though it is an often-discussed optical degradation pattern. Meanwhile, our experiments also include cases with pronounced chromatic aberration, as in Figure 5, where our method effectively suppresses purple fringing caused by chromatic aberration. These evidences indicate that although chromatic aberration is not considered in data sampling, it does not affect the model's performance in this respect.

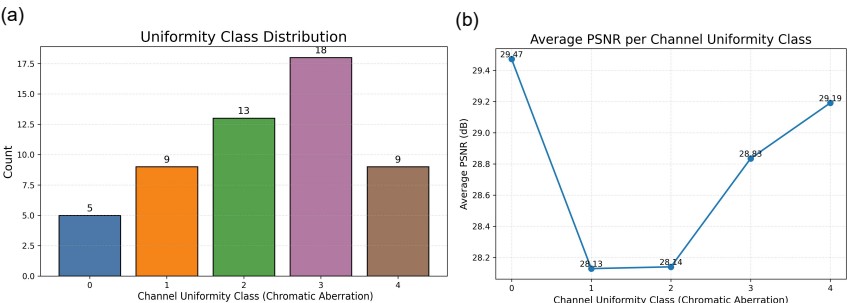

Figure 10: Evidence for omitting the chromatic aberration as the optical degradation classification criteria. (a) Distribution of different levels of chromatic aberrations in AODLibpro `Test`. (b) Performance (PSNR) of FoundCAC under different chromatic aberration levels.

Then, because the configuration of the EAOD algorithm does not include settings for a cemented doublet structure, the optimization imposes no strong constraint on chromatic aberration, which causes many generated samples to exhibit noticeable chromatic aberration. Figure 11 shows the distribution of chromatic aberration severity in our AODLibpro `Train`, from which it can be seen that a considerable portion of the training samples reveal obvious chromatic aberration. We believe that these chromatically degraded samples in the training set endow the model with the ability to handle chromatic aberration, enabling it to effectively address purple fringing across the test cases.

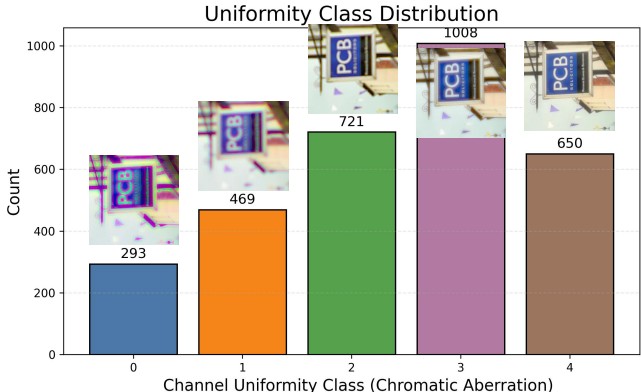

Figure 11: Distribution of different chromatic aberration levels in AODLibpro `Train`. Visualization cases for each level are also provided.

# E DETAILS FOR THE FOUNDCAC MODEL

## E.1 DETAILS FOR PSF-VQVAE

The encoder and decoder of PSF-VQVAE each use 3 groups of ResBlocks, with corresponding downsampling or upsampling convolutions in every group. The latent space for representation and feature processing operates at $1/8$ of the input resolution and has $n_z = 256$ channels. The structures of the encoder and decoder and the latent space feature dimensionality apply equally to the regularizer and FoundCAC, so we do not elaborate further. The codebook contains $K = 1024$ code vectors with a feature dimension of 512. During feature matching, two additional $1 \times 1$ convolution layers are applied before and after matching to handle feature dimension mismatches. These designs and settings follow Chen et al. (2022a); Esser et al. (2021).

## E.2 DETAILS FOR THE PSF-CONDITIONED REGULARIZER

For the PSF-conditioned fusion module $F_{PSF}$, we adopt Spatial Feature Transformation (SFT) followed by a cross attention module to generate the optical degradation image from a clear input conditioned on PSF features that encode spatial variation patterns. This fusion form allows the two features to interact over the full spatial domain and enables PSF features to spatially modulate intermediate image features, achieving the optical degradation modeling we need. We do not use a deep fusion backbone as in the aberration correction stage, because the optcial degradation process does not involve solving an ill-posed problem and therefore does not require many parameters or modules for deep fusion. In addition, since the regularizer acts as a constraint throughout training, excessive modules would increase the training cost. While richer regularizer architectures and fusion variants may further strengthen the constraint, our goal here is to verify that the regularizer helps learn a useful PSF latent representation, so we leave broader design exploration to future work.

## E.3 DETAILS FOR ABERRATION CORRECTION STAGE

For FoundCAC (RRDB), we use 6 RRDB layers from Wang et al. (2018) as the deep fusion backbone. For FoundCAC (Swin), we adopt 4 RSTB layers in Liang et al. (2021). For the RRDB layer, $num\_grow\_chn$ is set to 128, corresponding to a feature channel dimension of 256. For the RSTB layer, we set $blk\_depth$, $num\_heads$, and $window\_size$ to 6, 8, and 8, respectively. Since our RSTB operates at $1/8$ input resolution, FoundCAC achieves faster inference than SwinIR, which uses RSTB at $1/4$ resolution. In $F_C$, we adopt concatenation as the first step because our goal is to demonstrate that LPR can effectively guide aberration correction, and concatenation followed by deep fusion is an intuitive approach. Exploring more efficient fusion strategies is also an interesting direction for future work.

# F DETAILED SETTINGS FOR MAIN EXPERIMENTS

## F.1 DETAILS ON IMAGING SIMULATION

For AODLibpro `Train`, AODLibpro `Test`, and *RealLens-Sim*, we obtain the paired optical degradation images corresponding to each lens design via imaging simulation. The imaging simulator from DeepLens (Yang et al., 2024) is adopted, considering its precise calculation of PSFs and simulation of optical degradation images by patch-wise convolution. Specifically, we feed the design files of each sample in these LensLibs (in Zemax or parameter table format) into DeepLens for ray tracing to compute PSFs at 64 fields of view and 31 sampled wavelengths in the visible band. We match the closest sensor from the sensor library ($4\mu m - 2K$, $8\mu m - 2K$, $12\mu m - 2K$, $16\mu m - 2K$) based on the image height, discretize and sample the PSFs according to the pixel size, and stack them across RGB channels according to the wavelength response characteristics, ultimately obtaining the PSF arrays for each FoV and channel used in the simulation. It is worth noting that we applied equivalent downsampling to sensor resolution and pixel size to facilitate the use of large-scale public $2K$ high-quality image datasets. Meanwhile, unifying the resolution also helps to systematically control variables to build a benchmark for exploring the aberration correction task. Finally, the computed PSFs are used to convert clear images into the corresponding optical degradation images via patch-wise convolution, while the pipeline also accounts for sensor ISP and noise, as in most

imaging simulation workflows (Chen et al., 2021a;b). For each lens in the 3 datasets, we will open source its Zemax design file, the computed PSF array, and the simulated paired optical degradation images.

## F.2 MOTIVATION FOR SETTING AODLIBPRO TEST

Previous studies on blind aberration correction have applied specific test data (Eboli et al., 2022; Gong et al., 2024; Jiang et al., 2024b), which makes it inconvenient to evaluate the performance of model paradigms. In this situation, the optical degradation distributions of test lenses vary widely and unpredictably across works, so evaluations often measure the joint effect of training data and the model paradigm rather than the paradigm's own ability to learn optical degradation from data. Therefore, with training fixed on AODLibpro `Train`, we propose constructing AODLibpro `Test` as a benchmark for evaluating aberration correction networks. This benchmark has the following advantages: i) lenses in `Train` and `Test` are non-overlapping samples drawn from the same EAOD-generated lens source, which ensures that optical degradation distributions in `Test` are independently unseen while keeping the domain gap moderate, thereby focusing on the paradigm's ability to learn optical degradation; ii) constructing `Test` also uses the hybrid sampling basis, yielding optical degradation distributions that are uniform across spatial variation patterns and severity, with no data bias, thus enabling reliable evaluation of comprehensive aberration correction performance. We believe this benchmark setup can promote exploration of model paradigms and help ensure that the validations of various model designs are effective. All results in Table 3,4,7 are obtained under this training and testing setup.

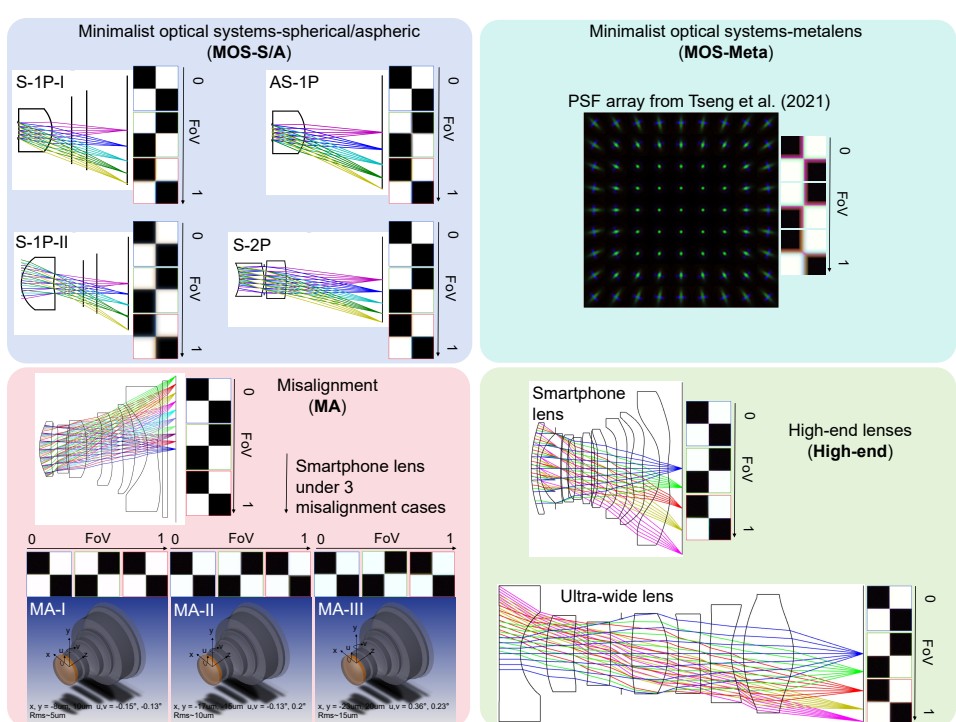

Figure 12: Illustration of lens designs, settings, and optical degradation patterns for *RealLens-Sim*.

## F.3 ILLUSTRATION OF LENS DESIGNS IN *RealLens-Sim*

Figure 12 shows the structures of the test lenses used in *RealLens-Sim* and example imaging results. The lenses are sourced from open source designs in the literature or from designs manually created by optical designers based on specifications in public patents or the needs of minimalist applications. In addition, we consider lens misalignment, a common but rarely addressed real-world factor that induces optical degradation. This typically occurs when the PSF size under the nominal lens design is below one pixel, that is, no optical degradation, but the decentering and tilt errors in manufacturing

and assembly lead to random unknown optical degradation in the final imaging results. We select a smartphone lens whose original design yields no sensor sampled optical degradation, set 3 groups of random decentering and tilt errors with increasing magnitude within its tolerance range, and then perform ray tracing to compute PSFs for imaging simulation.

Unlike AODLibpro `Test`, which provides a comprehensive evaluation in terms of optical degradation severity and spatial variation patterns in the imaging results, *RealLens-Sim* aims to provide test data from the perspective of lens types across real-world application scenarios, reflecting the practicality of the overall blind aberration correction framework and assessing the combined performance of training data and model paradigm. Admittedly, lens design cases in the real world are innumerable, and many are not open source, so one cannot include them all in the tests. Nevertheless, the lens types, application scenarios, and optical degradation distribution types covered by *RealLens-Sim* are the broadest among known open source data, allowing it to serve as a strong evaluation benchmark for assessing the generalization of blind aberration correction methods. In addition, because these lenses are manually designed by optical designers, they are out of domain relative to the LensLib generated by AOD methods, which can provide a fairer evaluation setting.

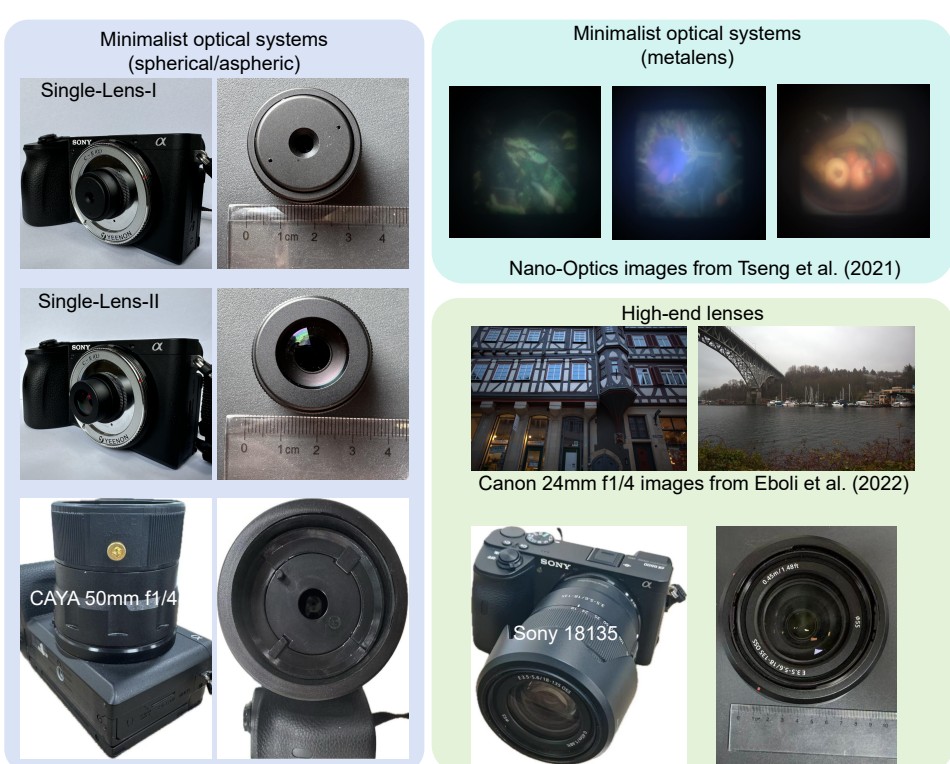

Figure 13: Schematic of the real snap setups for each lens in *RealLens-Snap*. For all devices whose setups are shown, we use images snapped in real-world scenes, while for the others we directly use their open-source images (Tseng et al., 2021; Eboli et al., 2022).

## F.4 ILLUSTRATION OF CAPTURE DETAILS FOR *RealLens-Snap*

We show the imaging setups used for our real-world captures in *RealLens-Snap* in Figure 13. In addition, the nano-optics data come from 3 open-source images in Tseng et al. (2021), and the $Canon$ $4mm$ $f1/4$ data come from two open-source images in Eboli et al. (2022). Quantitative evaluation on *RealLens-Snap* directly reflects the potential of blind aberration correction to improve image quality on real-world terminals. Given the difficulty of collecting real-world lenses under various applications for shooting, we strive to construct *RealLens-Snap* covering commonly used scenarios for blind aberration correction, such as minimalist optical systems, high-end photographic equipment, and metalens imaging. The selected systems exhibit distinct optical degradation distributions, enabling a more comprehensive evaluation of the representative methods. To the best of our knowl-

edge, we are the first work of blind aberration correction whose evaluation simultaneously covers minimalist optical systems with severe aberrations and high-end lenses with mild aberrations. We also hope to continuously collect more real snapped images with optical degradation in future work to broaden application scenarios.

### F.5 DETAILS FOR COMPETING BLIND ABERRATION CORRECTION METHODS

For the fast two-step method (Eboli et al., 2022), since we are dealing with spatially varying optical degradation, we process the optical degradation images using $256 \times 256$ tiles with 128 overlap, and keep all other settings the same as the defaults in its open-source code. Regarding the choice of an open-source, pretrained universal IR model for comparison, we consider it a primary option for users without an optical background to handle unknown optical degradation, because such methods aim to use highly generalizable large models to address real-world unknown degradations. We directly load its pretrained weights (Zhang et al., 2024) to process our optical degradation data. For all LensLib-PT methods (Gong et al., 2024; Jiang et al., 2024c; Côté et al., 2021; Jiang et al., 2024b) on the data side, we generate paired optical degradation images using the same pipeline as AODLibpro based on each LensLib's PSF arrays. Considering the inconsistency in the number of lenses across different Lenslibs, to ensure a fair comparison, we keep the total number of training images identical by changing the number of GT images degraded per lens. For the method in Gong et al. (2024), although the proposed model paradigm is insightful, we use only its LensLib ZEBASELib to train SwinIR for comparison rather than the proposed model because it is not open-sourced. For the pipelines in Jiang et al. (2024c) and Côté et al. (2021), since they provide only the ideas for constructing a LensLib and do not involve aberration correction model design, we likewise use only the LensLibs built following their ideas and use SwinIR as the model. For the SwinIR used as the network, we adopt the same $\mathcal{L}_C$ as FoundCAC (L1 loss and perceptual loss), with a batch size of 16 and $200K$ training iterations.

### F.6 DETAILS FOR TRAINING COMPETING ABERRATION CORRECTION NETWORKS

For all networks compared in Table 3, we use their architectures and retrain them on AODLibpro `Train`. To fully exploit the capability of each method, we adopt the official training configurations. Moreover, to ensure fairness in perceptual metrics, all methods are additionally trained with perceptual loss.

### F.7 DETAILS FOR TABLE 4, 7, AND 8.

Figure 14 together with Figure 7, show the model paradigms of the compared PSF representation methods in Table 4. For a fair comparison, the network architectures of the modules, the losses used in the VQVAE, and the aberration correction and the regularizer losses involved in these paradigms are all kept consistent with those in the final LPR. We also omit the illustration of PSF-VQVAE+Regularizer (Both applied), because this pipeline only adds a aberration correction network for supervision on top of LPR pre-training and applies that network as initialization in the aberration correction stage. Meanwhile, the baseline models in Table 7 and 8 are the first model paradigm shown in Figure 7.

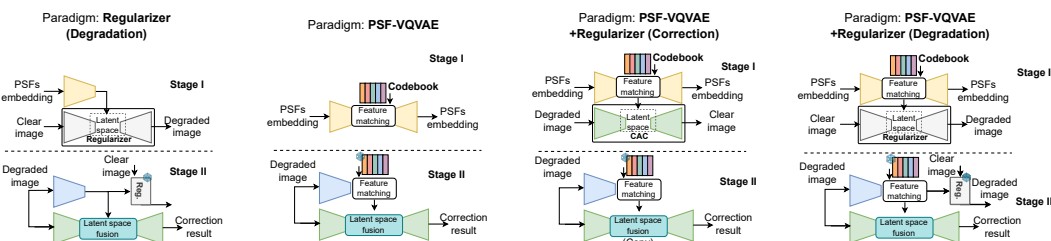

Figure 14: Illustration of the model paradigms in Table 4. Some of the paradigms have been shown in Figure 7, which are omitted here (baseline, GT-PSFs-guided, PSFs prediction, and PSFs feature prediction).

# G   MORE EXPERIMENTAL RESULTS

## G.1   ADDITIONAL RESULTS FOR THE EVALUATION ON EACH LENS OF *RealLens-Sim*

Table 10 reports per-lens results for each method as a complement to Table 2. Figure 15, 16, 17, and 18 shows qualitative correction results of representative methods on selected *RealLens-Sim* lenses. On lenses in MOS with more severe aberrations, when other methods still show residual optical degradation, OmniLens++ produces relatively clearer results without introducing false details and is close to the GT. For milder aberrations such as MA and high-end lenses, OmniLens++ can further correct aberrations to improve image quality while avoiding over-sharpening, yielding more natural results. These additional results highlight the good generalization of OmniLens++.

Table 10: Per-lens results for competing blind lens aberration correction methods on *RealLens-Sim*.

| Method | S-1P-I | | | S-1P-II | | | AS-1P | | | S-2P | | | Nano-Optics | | |
|---|---|---|---|---|---|---|---|---|---|---|---|---|---|---|---|
| | PSNR | SSIM | LPIPS | PSNR | SSIM | LPIPS | PSNR | SSIM | LPIPS | PSNR | SSIM | LPIPS | PSNR | SSIM | LPIPS |
| Fast two-step | 20.48 | 0.747 | 0.2783 | 18.87 | 0.681 | 0.4133 | 22.50 | 0.696 | 0.3077 | 20.90 | 0.758 | 0.2514 | 21.58 | 0.655 | 0.4488 |
| Universal IR | 20.05 | 0.761 | 0.2670 | 18.92 | 0.726 | 0.2698 | 22.00 | 0.734 | 0.2742 | 20.39 | 0.765 | 0.2351 | 21.36 | 0.687 | 0.4229 |
| ZEBASELib-PT | 23.84 | 0.813 | 0.2695 | 21.59 | 0.752 | 0.3736 | 22.26 | 0.767 | 0.3077 | 24.67 | 0.831 | 0.2369 | 18.14 | 0.679 | 0.4931 |
| ZernikeLib-PT | 26.39 | 0.855 | 0.1548 | 23.01 | 0.791 | 0.2534 | 23.61 | 0.779 | 0.1840 | 24.62 | 0.865 | 0.1442 | 20.50 | 0.707 | 0.3806 |
| AODLib-LensNet-PT | 23.70 | 0.826 | 0.2572 | 21.30 | 0.751 | 0.3978 | 22.48 | 0.768 | 0.3002 | 24.91 | 0.841 | 0.2194 | 18.56 | 0.678 | 0.4993 |
| AODLib-EAOD-PT | 27.83 | 0.880 | 0.1354 | 25.77 | 0.832 | 0.2002 | 24.29 | 0.803 | 0.1828 | 29.00 | 0.898 | 0.1202 | 20.10 | 0.740 | 0.3805 |
| AODLibpro-PT (SwinIR) | 27.30 | 0.877 | 0.1442 | 25.73 | 0.835 | 0.1923 | 26.79 | 0.837 | 0.1558 | 28.69 | 0.894 | 0.1242 | 22.02 | 0.754 | 0.3391 |
| **AODLibpro-PT (FoundCAC)** | 27.54 | 0.880 | 0.1396 | 25.63 | 0.840 | 0.1773 | 27.24 | 0.841 | 0.1466 | 27.96 | 0.896 | 0.1226 | 23.32 | 0.769 | 0.3145 |

| Method | MA-I | | | MA-II | | | MA-III | | | Ultra-wide lens | | | Smartphone lens | | |
|---|---|---|---|---|---|---|---|---|---|---|---|---|---|---|---|
| | PSNR | SSIM | LPIPS | PSNR | SSIM | LPIPS | PSNR | SSIM | LPIPS | PSNR | SSIM | LPIPS | PSNR | SSIM | LPIPS |
| Fast two-step | 28.05 | 0.789 | 0.1708 | 26.54 | 0.742 | 0.1796 | 26.81 | 0.771 | 0.1930 | 28.23 | 0.828 | 0.1710 | 27.97 | 0.825 | 0.1725 |
| Universal IR | 27.96 | 0.822 | 0.1699 | 27.40 | 0.805 | 0.1776 | 25.34 | 0.770 | 0.1864 | 29.33 | 0.851 | 0.1464 | 30.06 | 0.864 | 0.1383 |
| ZEBASELib-PT | 26.66 | 0.855 | 0.1071 | 26.53 | 0.846 | 0.1232 | 24.55 | 0.809 | 0.1518 | 27.80 | 0.898 | 0.1123 | 28.52 | 0.905 | 0.0982 |
| ZernikeLib-PT | 26.47 | 0.881 | 0.1130 | 25.02 | 0.833 | 0.1173 | 24.01 | 0.835 | 0.1440 | 26.58 | 0.897 | 0.0929 | 27.87 | 0.900 | 0.0934 |
| AODLib-LensNet-PT | 27.12 | 0.886 | 0.1024 | 25.90 | 0.840 | 0.1185 | 24.87 | 0.847 | 0.1464 | 27.33 | 0.901 | 0.1041 | 27.66 | 0.908 | 0.0912 |
| AODLib-EAOD-PT | 26.80 | 0.892 | 0.0939 | 26.41 | 0.855 | 0.0932 | 24.86 | 0.836 | 0.1174 | 27.64 | 0.901 | 0.0968 | 27.80 | 0.906 | 0.0938 |
| AODLibpro-PT (SwinIR) | 27.44 | 0.895 | 0.0968 | 26.48 | 0.858 | 0.0951 | 25.51 | 0.840 | 0.1186 | 27.47 | 0.902 | 0.1005 | 27.41 | 0.906 | 0.0979 |
| **AODLibpro-PT (FoundCAC)** | 28.58 | 0.896 | 0.0917 | 27.50 | 0.864 | 0.0927 | 26.24 | 0.838 | 0.1101 | 28.76 | 0.902 | 0.0981 | 28.86 | 0.906 | 0.0968 |

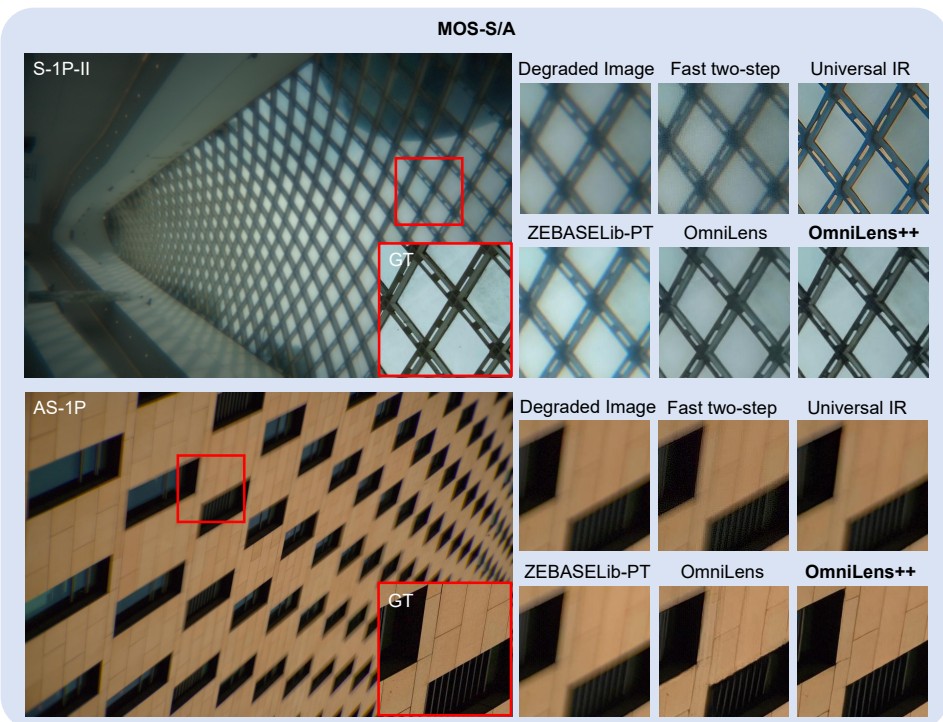

Figure 15: Visual comparison on MOS-S/A. S-1P-II and AS-1P are selected as the representative spherical and aspheric lenses for their distinct optical degradation patterns.

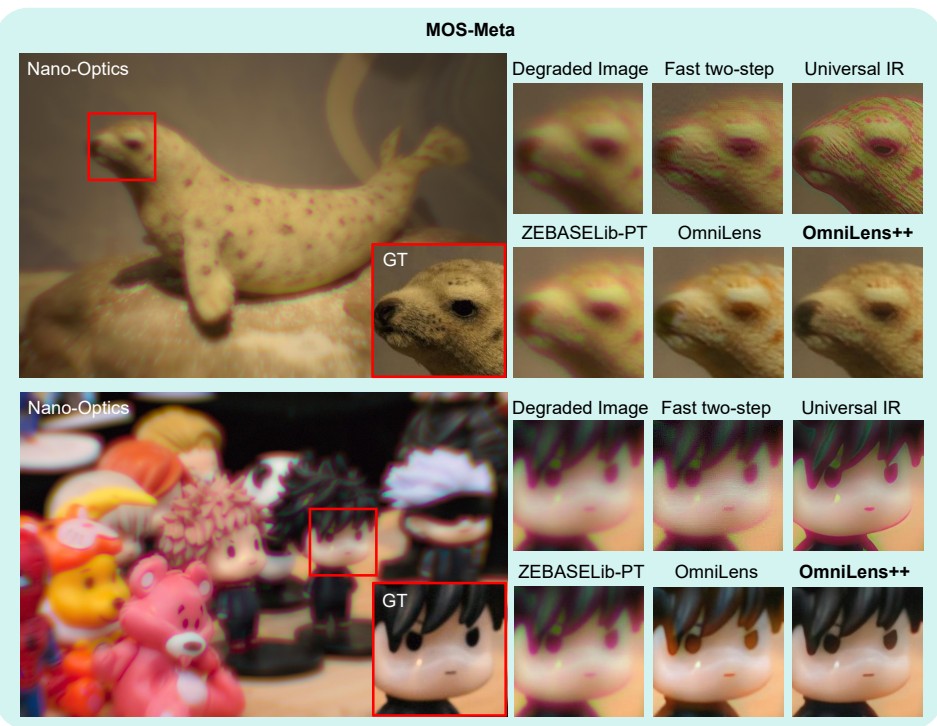

Figure 16: Visual comparison on MOS-Meta.

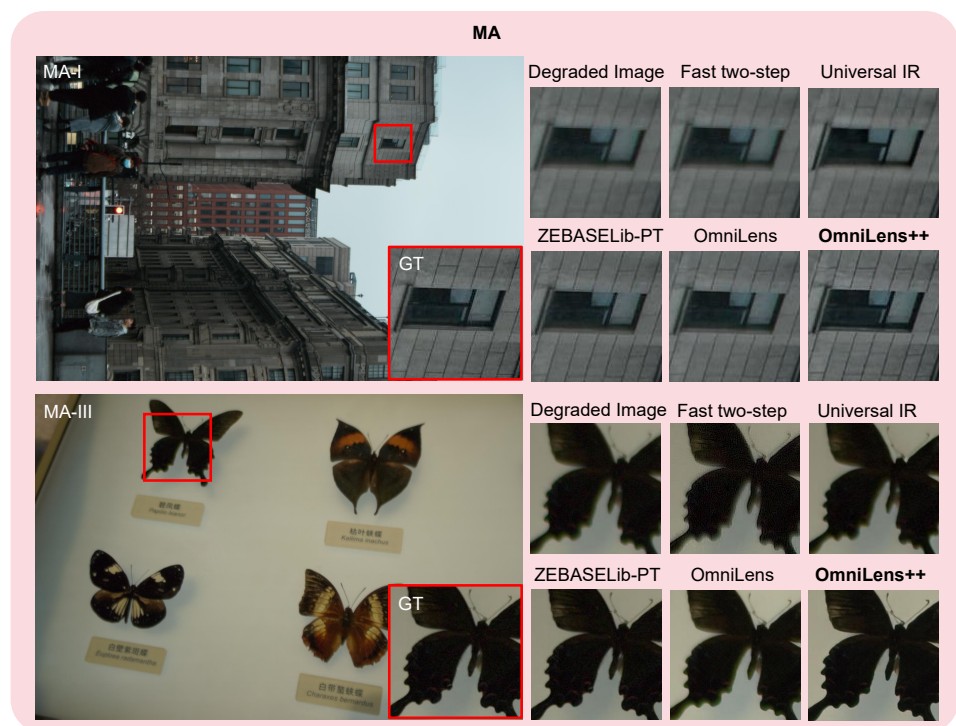

Figure 17: Visual comparison on MA. We present results under the minimum misalignment (MA-I) and maximum alignment (MA-III) settings.

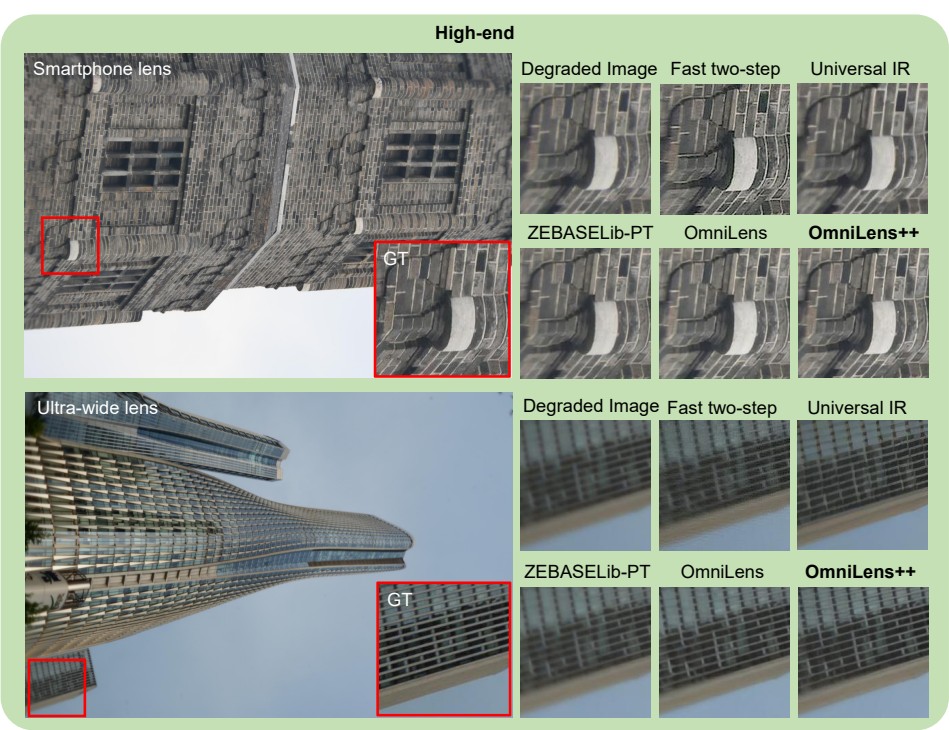

Figure 18: Visual comparison on High-end.

## G.2 ADDITIONAL ANALYSIS FOR SUPPLEMENTED SPECIFICATIONS.

As shown in Figure 19, using Spatial-Class, we visualize the optical degradation distributions of lens-source samples generated by EAOD under the baseline specifications, with aspheric surface added, and with both aspheric surface (A.S.) and image plane perturbation (I.P.P.) added. The results are consistent with our motivation for introducing these factors, namely that both specifications lead the optimized lenses to exhibit new optical degradation patterns. Specifically, adding A.S. yields more samples with severe optical degradation because the increased number of optimization parameters makes optimization more difficult, while adding image-plane perturbations yields more Spatial-Class "2" and "3" samples because shifting the image plane position changes the in-focus field.

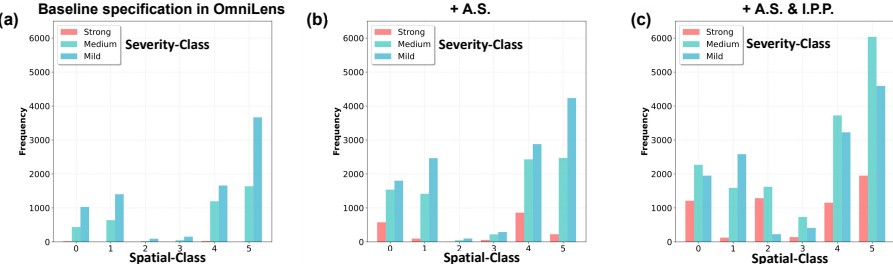

Figure 19: Optical degradation distributions of the lens source generated under different specification settings.

## G.3 ADDITIONAL ABLATIONS ON CODEBOOK SIZE

We sweep the codebook size $K$ to test whether allocating more entries for key latent PSF features benefits LPR. Table 11 shows no meaningful gains for FoundCAC from larger $K$. For the current PSF diversity in AODLibpro, $K = 1024$ appears sufficient to capture latent optical priors, whereas larger $K$ likely accumulates ineffective codes, complicates matching, and yields less reliable PSF guidance. A promising direction is to devise training strategies for larger codebooks that learn more effective degradation representations.

Table 11: Ablations on codebook size in LPR

| $K$ | PSNR | LPIPS |
|---|---|---|
| 512 | 28.53 | 0.1301 |
| 2048 | 28.78 | 0.1297 |
| **1024** | 28.67 | 0.1277 |

## G.4 VISUALIZATION OF THE LEARNED PSF REPRESENTATION

To understand why LPR provides better guidance for aberration correction than the PSF representations compared in Table 4, we visualize in Figure 20 the predicted PSF features under each representation before feature fusion. For two optical degradation patterns with completely different distributions, one spatially non-uniform with moderate overall severity and one more spatially uniform with strong overall severity, LPR shows clear differences between the predicted PSF features and can partially reflect their intuitive optical degradation patterns. Specifically, for S-1P-I, the attention regions of the PSF features predicted by LPR are mostly located at the peripheral FoVs, whereas for S-1P-II, the attention regions are almost global. These match the optical degradation patterns exhibited in the degraded image, indicating that LPR effectively learns optical degradation priors, which other representations cannot achieve. In PSF-VQVAE, the attention distribution is very sparse and lacks clear regularity; while in the regularizer, the PSF features contain more scene details, and the attention is almost the same across different optical degradation patterns, indicating that it fails to decouple the optical degradation priors encoded by PSFs. These pieces of evidence

further demonstrate the superiority of LPR in combining the two. In addition, even when directly supervising the prediction with GT PSF features, the predicted PSF features can hardly intuitively reflect the optical degradation pattern, although the differences across optical degradation cases are better than only applying the regularizer.

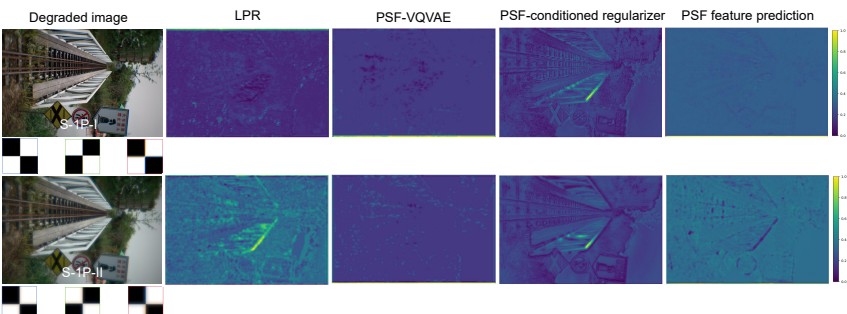

Figure 20: Visualization of the predicted latent PSF features. We use the PSF feature map before fusion to visualize the guidance of different PSF representations. The input optical degradation image and an example of its optical degradation distribution are shown in the first row.

### G.5 NUMERICAL EVALUATION ON *RealLens-Snap*

To more fairly and comprehensively evaluate the performance of representative blind aberration correction pipelines on real-snapped data *RealLens-Snap*, we conduct quantitative evaluation using common non-reference image quality metrics CLIPIQA (Wang et al., 2023), NIQE (Mittal et al., 2013), and MANIQA (Yang et al., 2022), with results shown in Table 12. Consistent with the results in Figure 5, even though these metrics exhibit some instability, the proposed OmniLens++ still performs better overall, highlighting its competitive generalization. Specifically, most of OmniLens++ results rank in the top 3 across these metrics (16/18), outperforming the second best method, Universal IR (14/18). Additionally, OmniLens++ shows no outliers in its results, that is, no cases with particularly poor performance. Since Universal IR uses S3Diff, a diffusion based generative model whose outputs exhibit unrealistic sharpening yet enjoy inherent advantages on these non-reference metrics (Zhang et al., 2024), these results indicate the favorable robustness and generalization of OmniLens++.

Table 12: Per-lens numerical evaluation for competing blind lens aberration correction methods on *RealLens-Snap*. The **first**, second, and *third* results are highlighted.

| Method | Single-Lens-I | | | Single-Lens-II | | | CAYA 50mm f1/4 | | |
|---|---|---|---|---|---|---|---|---|---|
| | CLIPIQA↑ | NIQE↓ | MANIQA↑ | CLIPIQA↑ | NIQE↓ | MANIQA↑ | CLIPIQA↑ | NIQE↓ | MANIQA↑ |
| Fast two-step | 0.341 | 3.903 | 0.218 | 0.341 | 4.907 | 0.227 | 0.324 | **3.835** | 0.180 |
| Universal IR | **0.470** | **3.548** | **0.310** | 0.407 | *5.575* | *0.282* | **0.411** | 3.843 | *0.266* |
| ZEBASELib-PT | 0.336 | 5.493 | 0.211 | 0.319 | 7.265 | 0.213 | 0.299 | 5.966 | 0.202 |
| OmniLens | *0.383* | 4.789 | 0.289 | *0.392* | 5.718 | **0.327** | *0.376* | 4.311 | **0.277** |
| **OmniLens++** | 0.398 | *4.061* | *0.272* | **0.456** | **4.792** | 0.310 | 0.393 | *3.929* | 0.273 |

| Method | Nano-Optics | | | Canon 24mm f1/4 | | | Sony 18135 | | |
|---|---|---|---|---|---|---|---|---|---|
| | CLIPIQA↑ | NIQE↓ | MANIQA↑ | CLIPIQA↑ | NIQE↓ | MANIQA↑ | CLIPIQA↑ | NIQE↓ | MANIQA↑ |
| Fast two-step | 0.342 | **7.013** | 0.231 | 0.469 | **3.292** | *0.343* | **0.560** | **2.826** | 0.303 |
| Universal IR | **0.389** | *8.172* | 0.263 | 0.433 | 3.388 | 0.332 | 0.489 | 3.050 | *0.317* |
| ZEBASELib-PT | *0.360* | 9.522 | **0.299** | 0.519 | 4.182 | 0.359 | 0.512 | 4.264 | 0.322 |
| OmniLens | 0.332 | 8.735 | 0.285 | *0.492* | 3.971 | **0.369** | 0.499 | 3.955 | **0.328** |
| **OmniLens++** | 0.370 | 8.126 | *0.266* | **0.538** | *3.790* | 0.339 | *0.500* | *3.395* | 0.315 |

### G.6 ADDITIONAL VISUAL RESULTS ON *RealLens-Snap*

To further demonstrate the effectiveness of our method, we present additional comparison results between the proposed OmniLens++ and competing approaches on real-world images captured with different lenses. Results on simple spherical and aspherical lenses are shown in Figures 21, 22, and 23, those on metalens are shown in Figure 24, and those on high-end lenses are shown in Figures 25 and 26. The proposed OmniLens++ consistently produces favorable correction results on various lens types, further demonstrating its zero-shot capability in handling diverse aberrations. Specifically, for MOS, OmniLens++ mitigates severe aberrations and delivers promising correction results where competing methods perform unsatisfactorily. For high-end DSLR lenses, it further alleviates residual aberrations, including chromatic aberration, while avoiding over-sharpening.

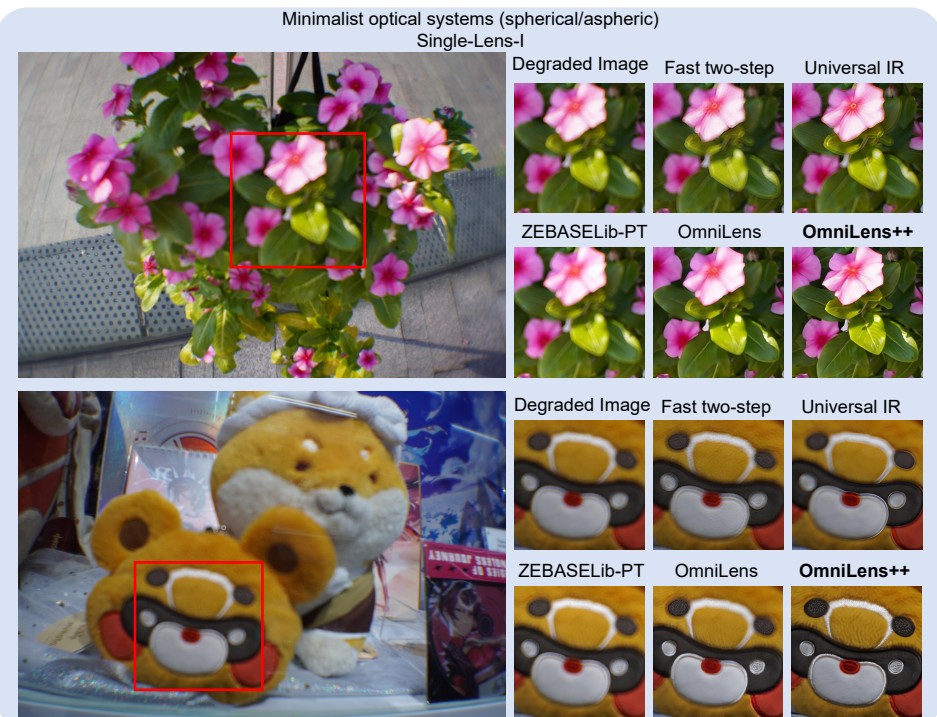

Figure 21: Visual comparison on Single-Lens-I.

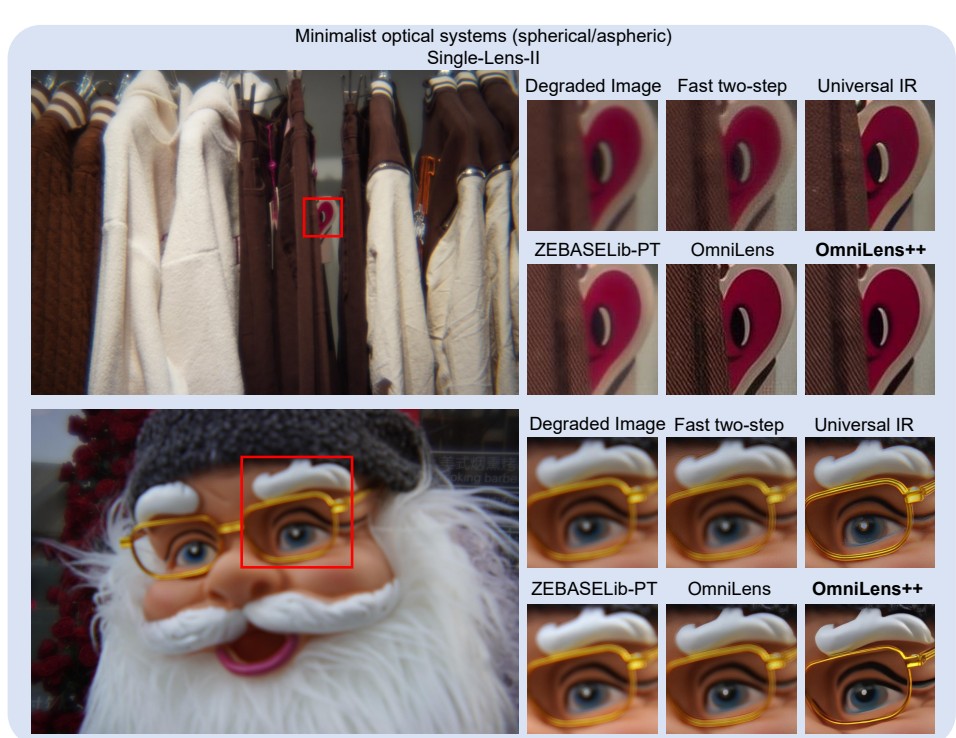

Figure 22: Visual comparison on Single-Lens-II.

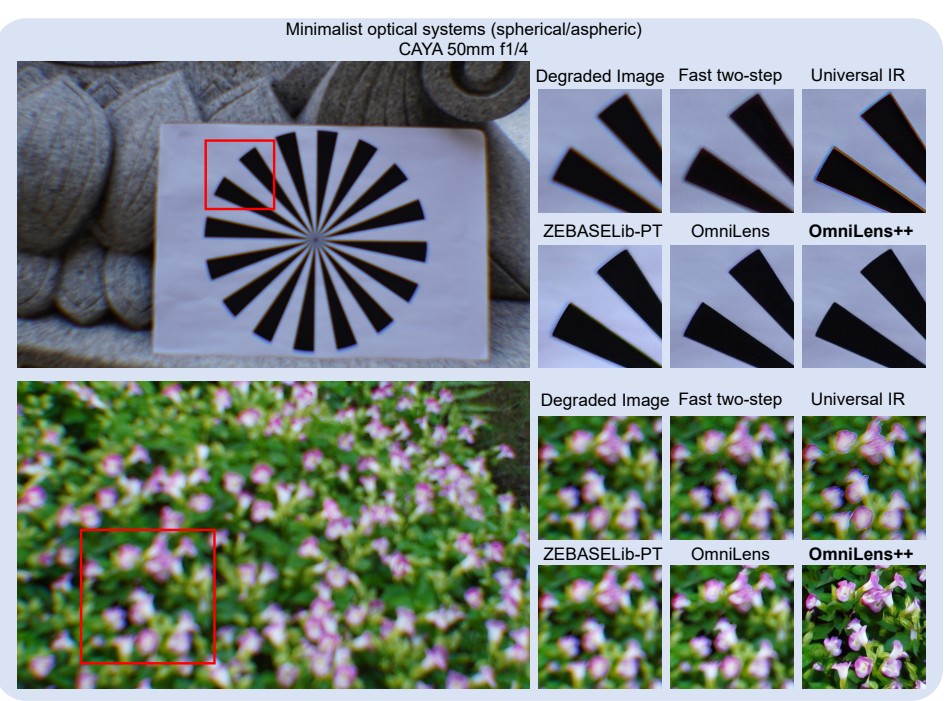

Figure 23: Visual comparison on $CAYA\ 50mm\ f1/4$.

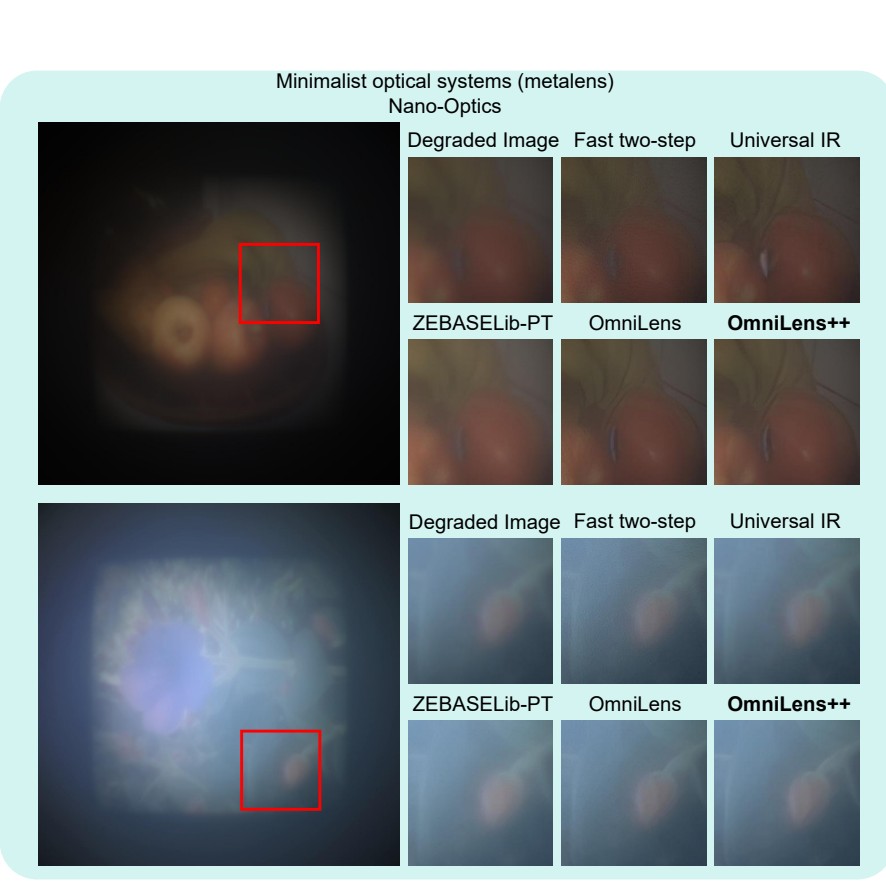

Figure 24: Visual comparison on Nano-Optics.

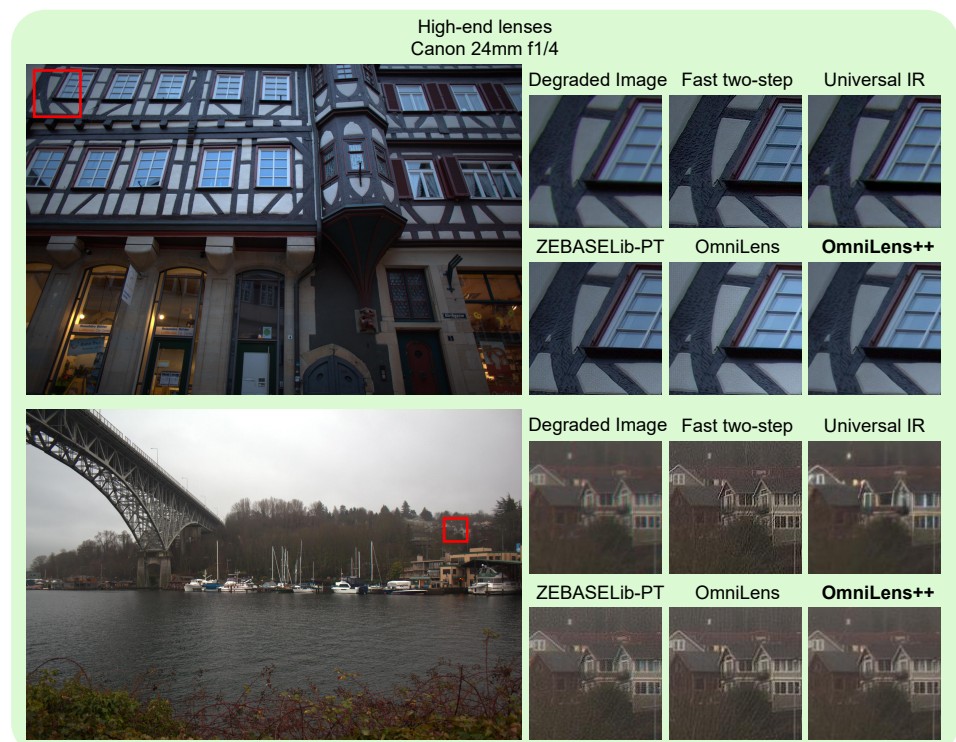

Figure 25: Visual comparison on $Canon\ 24mm\ f1/4$.

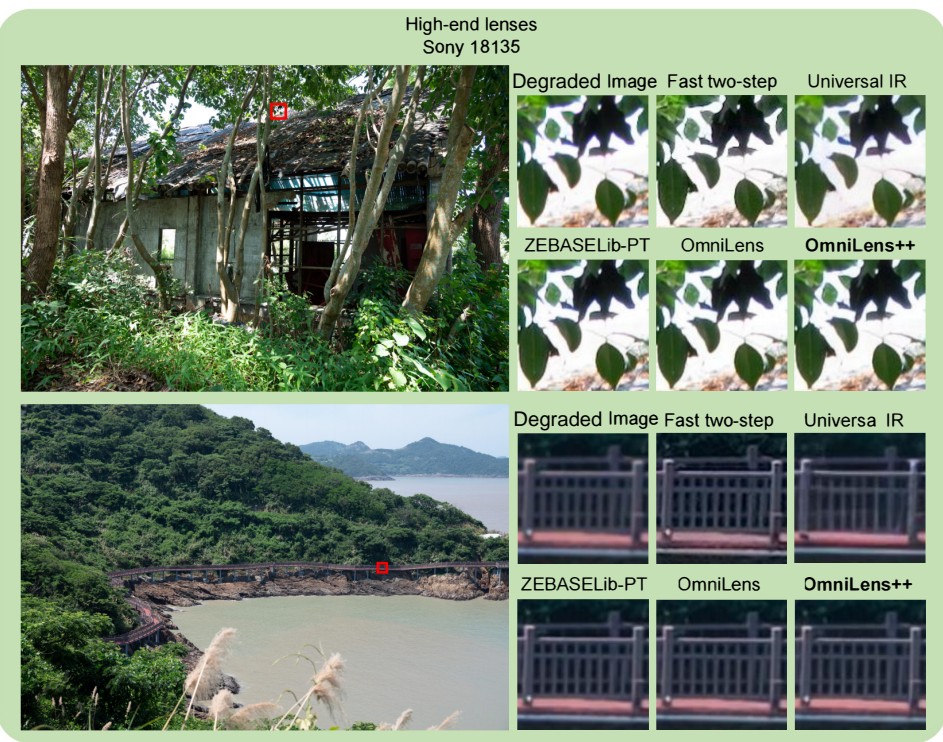

Figure 26: Visual comparison on $Sony\ 18135$.

