# OpenReview forum: "OmniLens++: Blind Lens Aberration Correction via Large LensLib Pre-Training and Latent PSF Representation"
_ICLR.cc/2026/Conference — Submitted to ICLR 2026_

### Official Review · Reviewer_6hkZ · 2025-10-28

**Soundness:** 2
**Presentation:** 3
**Contribution:** 2
**Rating:** 4
**Confidence:** 5

**Summary:**

This paper presents an image restoration method for optical aberration removal that generalizes well across diverse optical lenses. By overfitting existing optical aberrations (PSFs) using a large-scale lens dataset, the network is exposed to a variety of aberrations, thereby improving generalization performance. The key improvement of this work compared to the prior arXiv paper (OmniLens) lies in its scalability. To achieve this, the paper expands the lens design specifications and introduces a latent representation of the PSF.

**Strengths:**

I agree that training an “all-in-one” network for various lens aberrations is an interesting academic exploration, but which requires a more comprehensive benchmark.

**Weaknesses:**

1. Regarding the paper's writing, two points need improvement:
  - The baseline for this paper is the arXiv paper “OmniLens: Towards Universal Lens Aberration Correction via LensLib-to-Specific Domain Adaptation.” However, the OmniLens framework is not clearly described in the main text. Readers may also be confused about whether there is an "OmniLens+" between OmniLens and OmniLens++.
  - There are too many uncommon abbreviations, for example, OD for optical degradation, LPR for latent PSF representation, CAC for computational aberration correction, and OIQ for optical image quality. These abbreviations are usually unnecessary and significantly increase reading difficulty.

2. Regarding the experiments:
  - The baseline (OmniLens) remains on arXiv, and this paper is a clear follow-up to it. Benchmarking against an un-peer-reviewed work is risky and may be hard to convince readers. The evaluation cases and dataset are highly customized by the authors, whereas a more comprehensive, standardized benchmark should be proposed.
  - Similar to Chen et al., this paper only considers plane object scenes. However, in real scenarios, different objects appear at different depths, introducing non-uniform defocus effects. The corresponding defocus datasets and simulation works should not be ignored. Examples include:
    - “Defocus Deblurring Using Dual-Pixel Data”
    - “Learning to Deblur Using Light Field Generated and Real Defocused Images”
    - “Aberration-Aware Depth-from-Focus”
    - “Efficient Depth- and Spatially-Varying Image Simulation for Defocus Deblur”

3. Debate on “all-in-one” image restoration:
  - I agree that training an “all-in-one” network for various lens aberrations is an interesting academic exploration, and there are related works on this topic for image denoising, motion deblurring, and deraining. However, different optical lenses can exhibit very different aberrations. Training on such a large-scale lens dataset may improve network performance on general camera lenses, but the results on metalenses (Figure 3) are not promising enough. In short, it is challenging to achieve zero-shot generalization to customized optics if they are not included in the dataset—this seems like a clear failure case for the overall idea of this paper. Examples include:
    - “Perspective-Aligned AT Mirror with Under-Display Camera”
    - "Removing Diffraction Image Artifacts in Under-Display Camera via Dynamic Skip Connection Networks"
  - In practice, different camera sensors have distinct noise profiles, which often require image restoration networks to be retrained or fine-tuned for different camera systems. As the dataset scale increases, overfitting all existing PSFs becomes significantly more difficult and demands larger network sizes and more complex architectures, posing challenges for deployment. In such cases, the practicality of an “all-in-one” image restoration network is marginal.

4. Regarding novelty:
  - The original idea of OmniLens is new, while scaling it up (e.g., by using aspherical surfaces, which is a standard approach) seems marginal.

**Questions:**

Please check weaknesses

---

> ### Author Response · Authors · 2025-11-20
> **Official Response by Authors -- Part 1**
>
> We would like to sincerely thank the reviewer for the detailed and constructive comments on our work. We take every comment seriously and hope these responses resolve your concerns, especially by helping you recognize the work’s novelty and significance, the completeness of the experimental validation loop, and the soundness of the evaluation benchmark. If there are any remaining questions, we are more than happy to address them.
>
> > **W1-1: Regarding the paper's writing, two points need improvement:**
> > **The baseline for this paper is the arXiv paper “OmniLens: Towards Universal Lens Aberration Correction via LensLib-to-Specific Domain Adaptation.” However, the OmniLens framework is not clearly described in the main text. Readers may also be confused about whether there is an "OmniLens+" between OmniLens and OmniLens++.**
>
> **We agree that** the OmniLens baseline should be better introduced, and **have made the following revisions** which can be found **in the revised paper**:
> * In L64-66, L68-69, and L72-73, we introduce the core ideas and key techniques of OmniLens and delineate the specific shortcomings of prior frameworks that our approach addresses.
> * The revised Figure 1 succinctly highlights the data and model challenges in the OmniLens framework that OmniLens++ resolves, serving as a clear comparison between OmniLens++ and OmniLens, where the two "+" are illustrated.
> * The supplemented Section 2 on Related Work offers a focused survey of lens aberration correction (L122–L143), outlining the strengths and limitations of the OmniLens baseline and other LensLib‑PT approaches, and clarifying the motivation behind OmniLens++. This added background should help readers transition smoothly into our idea.
>
> > **W1-2: There are too many uncommon abbreviations, for example, OD for optical degradation, LPR for latent PSF representation, CAC for computational aberration correction, and OIQ for optical image quality. These abbreviations are usually unnecessary and significantly increase reading difficulty.**
>
> We appreciate the suggestion and **have removed the unnecessary abbreviations OD and CAC in the latest revision** to improve readability. We retain LPR and OIQ because they denote our proposed method. Additionally, Table 9 in Appendix.B lists the full expansions of all abbreviations used in the paper to facilitate smoother reading.

---

> ### Author Response · Authors · 2025-11-20
> **Official Response by Authors -- Part 2**
>
> > **W2-1: Regarding the experiments:**
> > **The baseline (OmniLens) remains on arXiv, and this paper is a clear follow-up to it. Benchmarking against an un-peer-reviewed work is risky and may be hard to convince readers. The evaluation cases and dataset are highly customized by the authors, whereas a more comprehensive, standardized benchmark should be proposed.**
>
> Thank you for your comments. We show, on several fronts, that **OmniLens++ employs a comprehensive experimental protocol and evaluation benchmark** that is independent of the OmniLens benchmark settings.
>
> * **The originality of OmniLens++.** We would like to clarify first that OmniLens++ benefits only the EAOD‑based batch generation concept from OmniLens. Our work centers on samlpling the final scalable LensLib and designing model paradigms to handle diverse aberrations, which are not covered in OmniLens. **The core ideas are enhanced by the revised Figure 1 and related analysis (L72-L87, L88-L102).**
> * **Comprehensive benchmark.** Our benchmark differs fundamentally from OmniLens. Whereas OmniLens relies on a few ad hoc lens cases with limited coverage, we introduce three comprehensive benchmarks: AODLibpro Test, RealLens-Sim, and RealLens-Snap. Given that evaluation data in this field varies from work to work, **our benchmarks are, to our knowledge, the most comprehensive, covering lenses from minimalist to high-end with diverse aberration patterns.** The benchmarks **will be made publicly available**. We have emphasized this contribution (L86-L87) and summarize each benchmark’s characteristics below.
>     1. **AODLibpro Test.** The implementation details are provided in L347-L350, and the motivation behind it is provided in Appendix.F.2 (L1034-L1049). With training fixed on AODLibpro Train, we propose constructing AODLibpro Test as a benchmark for evaluating aberration correction networks. This benchmark uses non overlapping train and test splits drawn from the same source, and the test set uniformly and comprehensively covers optical degradation patterns.
>     2. **RealLens-Sim.** Appendix.F.3 (L1078-L1097) and Figure 11 illustrates its construction details and motivation. This setting aims to provide test data (simulated) from the perspective of lens types across real-world application scenarios, reflecting the practicality of the overall blind aberration correction framework and assessing the combined performance of training data and model paradigm.
>     3. **RealLens-Snap.** Appendix.F.4 (L1127-L1140) and Figure 12 illustrates its construction details and motivation. This is the real-snapped version of RealLens-Sim. Given the difficulty of collecting real-world lenses under various applications for shooting, we strive to construct RealLens-Snap covering commonly used scenarios for blind aberration correction, such as minimalist optical systems, high-end photographic equipment, and metalens imaging.
> * **Comarison methods.** We clarify that **our work is not simply benchmarking against OmniLens**. We evaluate representative state of the art baselines in lens aberration correction and applicable methods from related areas on our comprehensive benchmark (Table 2, Figure4, L372-L377). At the paradigm level, we compare strong image restoration and aberration correction architectures (Table 3, L429-L445) and additionally compares alternative PSF representations(Table 4, L457-L465). These rigorous studies substantiate the method’s effectiveness and support our claims.

---

> ### Author Response · Authors · 2025-11-20
> **Official Response by Authors -- Part 3**
>
> > **W2-2: Similar to Chen et al., this paper only considers plane object scenes. However, in real scenarios, different objects appear at different depths, introducing non-uniform defocus effects. The corresponding defocus datasets and simulation works should not be ignored. Examples include: ...**
>
> Thank you for the thoughtful suggestion. **We agree that** extending blind aberration correction pipeline to depth-aware PSF blur is a significant future work direction, and **we have added a discussion of your suggested references in Section 5 (L533–L539).**
> **Additional clarification:**
> * Our work targets at building a universal model that robustly handles PSF induced lens degradations with diverse spatial variation patterns and severities, which is both academically and practically valuable. Hence the evaluation emphasizing coverage across optical degradation patterns **of different lens types** is sufficient to support this research scope.
> * **Defocus blur is already covered**. The image plane perturbation in Figure 2 (a) models defocus, ensuring that AODLibpro‑Train and AODLibpro‑Test contain representative defocus samples.
> * **Several real-snapped cases exhibit depth-aware PSF blur from depth of field**, for example the first row on the left of Figure 4 and examples in Figures 20 to 22. Although the simulator in training set assuming a planar object distance did not model depth-aware PSFs, incorporating the above mentioned defocus specifications enables OmniLens++ to outperform competing methods on depth-aware PSF blur.
> * Most prior aberration correction studies [1-3] simulate with a planar object distance assumption. While depth and depth of field are important for practical deployment, this simplification is adequate for investigating lens type dependent optical degradation patterns.
>
> [1] Gong, Jin, et al. "A physics-informed low-rank deep neural network for blind and universal lens aberration correction." Proceedings of the IEEE/CVF Conference on Computer Vision and Pattern Recognition. 2024.
> [2] Chen, Liqun, et al. "A Physics-Informed Blur Learning Framework for Imaging Systems." Proceedings of the Computer Vision and Pattern Recognition Conference. 2025.
> [3] Li, Xiu, et al. "Universal and flexible optical aberration correction using deep-prior based deconvolution." Proceedings of the IEEE/CVF International Conference on Computer Vision. 2021.

---

> ### Author Response · Authors · 2025-11-20
> **Official Response by Authors -- Part 4**
>
> > **W3-1: Debate on “all-in-one” image restoration:**
> > **...However, different optical lenses can exhibit very different aberrations. Training on such a large-scale lens dataset may improve network performance on general camera lenses, but the results on metalenses (Figure 3) are not promising enough. In short, it is challenging to achieve zero-shot generalization to customized optics if they are not included in the dataset—this seems like a clear failure case for the overall idea of this paper. Examples include:...**
>
> Thank you for your insightful comment.
> * **We agree that** zero‑shot generalization of universal models depends strongly on data coverage, and out‑of‑distribution cases may lead to failure cases, as seen with the mentioned metalens.
> *  However, our experiments show that expanding design specifications during data generation improves performance on the corresponding cases; for example, adding aspheric surface enables OmniLens++ to handle aspheric lens cases. **This scalability underscores our contribution**, since the proposed novel sampling basis will benefit accommodation of more lens types and specifications to produce broader and more effective training data in the future work.
> * In the metalens case of Figure 4, aberration‑induced blur and color fringing are corrected, while residual artifacts stem from stray light under capture environment that was not modeled, and thus lies outside the method’s scope. The suggested work of under‑display imaging is a similar capture factor. Along with the mentioned depth of field above, **these optics‑coupled degradations are important for deployment and have been discussed in Section 5 (L533-L539)**.
> * These factors are beyond this paper’s scope. Here we focus on lens‑design‑induced aberrations, and our experiments sufficiently validate the method on this problem.
>
> > **W3-2: In practice, different camera sensors have distinct noise profiles, which often require image restoration networks to be retrained or fine-tuned for different camera systems. As the dataset scale increases, overfitting all existing PSFs becomes significantly more difficult and demands larger network sizes and more complex architectures, posing challenges for deployment. In such cases, the practicality of an “all-in-one” image restoration network is marginal.**
>
> We appreciate your concerns on the "all-in-one" paradigm. We hope the following justifications clarify the practicality of OmniLens++.
>
> * In practice, we can train a universal model for each common sensor style to perform aberration correction, thereby addressing adaptation across different sensors.
> * Your opinion on deployment is well taken. Research on "all-in-one" model remains nascent, and its practical utility is constrained by the trade‑off between model capacity and available computational overhead. However, **for blind aberration correction, this direction is worthwhile.** The target deployment is post‑processing software for non‑expert users, where computational constraints are relatively relaxed. It is therefore meaningful to scale both data and model capacity to push zero‑shot performance.
> * For mobile optical deployments, a practical approach is to pretrain a universal model and finetune it for the target device, which helps balance performance and computational overhead and is initially verified in OmniLens. This direction has been discussed in Section 5 (L538-L539).

---

> > ### Comment · Reviewer_6hkZ · 2025-11-24
> > **Follow-up review**
> >
> > I agree that "for blind aberration correction, this direction (OmniLens and OmniLens++) is worthwhile" and it is possible to provide a practical pipeline (degradation function) for all-in-one image aberration correction. However, I also hold the opinion that including stray light simulation and various degradation models can further improve the contribution of this work and better distinguish it from the original OmniLens work.
> >
> > I could not find the revised manuscript, but I believe my concerns can be well addressed in the revised version.
> >
> > I have improved my scores on this paper. Now the decision is on the AC. If the paper is not accepted, I would suggest the authors improve the paper on the points we discussed and make it a practical work for both academia and industry.

---

> ### Author Response · Authors · 2025-11-20
> **Official Response by Authors -- Part 5**
>
> > **W4: Regarding novelty: The original idea of OmniLens is new, while scaling it up (e.g., by using aspherical surfaces, which is a standard approach) seems marginal.**
>
> We appreciate your concerns and **have revised the manuscript to more clearly articulate our novelty and contributions.** Below we also offer additional justifications of the work’s novelty:
> * **Overall originality.** OmniLens++ benefits only the EAOD‑based batch generation concept from OmniLens, tackling deeper topics: given large scale generated lens samples, how to curate training data with comprehensive distributions that enhance generalization, and how to guide models to learn from these PSF-convolution-induced degradations. These aspects are not covered by OmniLens but pivotal to model performance. The improvements of OmniLens++ over OmniLens are significant (Figure 1, Figure 4, Figure 14-17, Figure 20-25, Table 1). We have **revised Figure 1** to foreground the problems addressed by OmniLens++ and **add a Related Work section (L120-L161)** to elaborate the background and motivation.
> * **Data construction.** The concrete pipeline for constructing AODLibpro, in particular the core lens quantification and sampling strategy, is entirely new. Our image quality based optical degradation quantification and hybrid sampling basis are the keys that give AODLibpro its scalability, **rather than simply scaling it up with additional specifications.** **The specific manuscript updates are as follows**:
>     1. In L79-L84, we have revised the AODLibpro exposition to foreground its core idea of quantifying optical degradation severity and spatial variation trends from an image quality perspective, underscoring its novelty.
>     2. The updated Figures 1 and 2 provide intuitive views of AODLibpro’s advantages and core technical pipeline.
>     3. In Section 3.2 (L221–L223, L238–L239), we also added specific motivation behind AODLibpro to further highlight its originality.
> * **Coupling between model design and data design.** LPR delivers effective PSF guidance in a blind manner and is both novel and well motivated. Beyond standalone ablations that validate the hybrid sampling basis (Table 6, Figure 5,  L480-L502) and LPR (Table 4, Table 7, L457-L465, L503-L511), we further study their interaction (Table 8, L517-L523). Results indicate that LPR can better leverage the scalability of AODLibpro and closes the loop between motivation and empirical evidence. We have also revised the presentations of the model side motivation (L96-L97, L263, L269-L271, L289-L290).

---

> ### Author Response · Authors · 2025-11-24
> **Follow up**
>
> Dear Reviewer 6hkZ:
>
> Thank the valuable comments on our paper, which provided insights that can help us revise our work.
>
> We have provided a response and a revised paper, hope they could address your concerns, especially on the originality of OmniLens++, the breadth of our benchmarking, and the soundness of the problem formulation.  Also, we would like to know if there are more concerns about the content of the paper. Your invaluable feedback and suggestions are greatly welcomed to help us better refine our work.
>
> Thank you again for your devotion to the review. If all the concerns have been successfully addressed, please consider raising the scores after this discussion phase.
>
> Best,
>
> Submission#3893 Authors

---

> ### Author Response · Authors · 2025-11-24
>
> Dear Reviewer 6hkZ:
>
> Thank you for your response!  We’re delighted to hear that our reply addressed your concerns and appreciate your recognition of our work. You can download the updated PDF by clicking the button next to the title at the top of the page. The file reflects our latest revision. We also appreciate your suggestions on including more degradation models in simulation. As discussed in Section 5, this is an important direction for future work to boost the practical use in industry of OmniLens++.
>
> Best,
>
> Submission#3893 Authors

---

### Official Review · Reviewer_Q7Qo · 2025-10-31

**Soundness:** 2
**Presentation:** 1
**Contribution:** 3
**Rating:** 4
**Confidence:** 2

**Summary:**

This work proposes OmniLens++, a framework for blind computational aberration correction build on Lens Library Pre-Training (lensLib-PT).
The authors claim two main contributions. First, they introduce a larger, better-balanced library (AODLibpro) that expands the design space with aspheric surfaces and image-plane perturbations to broaden degradation diversity.
Second, they propose a Latent PSF Representation (LPR) that injects PSF knowledge into a blind correction pipeline.
Using LPR as guidance, the authors train a foundational aberration-correction model (FoundCAC).
Across RealLens-Sim and real photographs, OmniLens++ delivers consistent improvements over prior LensLib-PT variants and strong deconvolution baselines.

**Strengths:**

Combining a broadened, more uniformly sampled synsthetic lens library with a learned latent prior is technically sound and novel.
The experimental validation is extensive, spanning diverse simulated aberration settings, and includes ablation studies showing that AODLibpro improves over earlier AODLib-EAOD settings and LPR guidance helps as data scale increases.

**Weaknesses:**

The presentation is needlessly hard to parse. The paper is overloaded with abbreviations (OD, CAC, ODN, OIQ, etc.), many of which describe concepts that could be expressed directly or formally. Optical degradation (OD) is essentially a forward operator in an inverse problem. Computational Aberration Correction could simply be described as image reconstruction and an Optical Degradation Network is, at its core, a neural network modeling the forward process. This naming density obscures rather than clarifies the contributions.
The method section is dense and very hard to follow. It introduces multiple interdependent modules (VQVAE, ODN, LPR, FoundCAC) in quick succession, often without intuitive explanation or guiding diagrams. Figures 1 and 2 are overly complex and fail to provide a clear overview of the system. They require significant prior knowledge of the text to decode. They are repeatedly cited as explanatory ("as shown in Figure 1 (a)...", "as shown in Figure 2 (a)...") when they in fact demand the textual explanation to be understood. Key design flows and data construction steps should be broken into simpler, sequential diagrams. The baseline OmniLens method is introduced too late in the text. Readers unfamiliar with the earlier work will struggle to contextualize what is actually new. The phrasing "constructing the large LensLib AODLibpro" reads as though the dataset already existed. Clarifying that AODLibpro is newly proposed and constructed in this work would strengthen the narrative. The writing, particularly in the methodology section, reads like a technical report rather than a scientific paper. Many sentences could be made clearer by explaining the motivation and intuition behind each modeling choice.

**Questions:**

- The Optical Image Quality (OIQ) metric is central to AODLibpro and the evaluation, but its motivation and validation are unclear. Could the authors justify why OIQ is needed for assessing blind lens aberration correction and show that it correlates better with perceptual or optical quality than PSNR or SSIM?
- The experiments seem simulation-heavy. Are there quantitative evaluations on real aberrated captures, not just qualitative examples?

---

> ### Author Response · Authors · 2025-11-20
> **Official Response by Authors -- Part 1**
>
> We would like to sincerely thank the reviewer for the useful comments on our work. To effectively respond to all your concerns, **we first provide all the modifications we have made on the presentation of the paper** following your suggestions.
>
> > **The presentation is needlessly hard to parse. The paper is overloaded with abbreviations (OD, CAC, ODN, OIQ, etc.), many of which describe concepts that could be expressed directly or formally. Optical degradation (OD) is essentially a forward operator in an inverse problem. Computational Aberration Correction could simply be described as image reconstruction and an Optical Degradation Network is, at its core, a neural network modeling the forward process. This naming density obscures rather than clarifies the contributions.**
>
> We appreciate the suggestion and **have removed the unnecessary abbreviations OD and CAC in the latest revision** to improve readability. We retain ODN and OIQ because they denote our proposed module and method, and their abbreviations facilitate concise notation and formula definitions. Additionally, Table 9 in Appendix.B lists the full expansions of all abbreviations used in the paper to facilitate smoother reading.
>
> > **The method section is dense and very hard to follow. It introduces multiple interdependent modules (VQVAE, ODN, LPR, FoundCAC) in quick succession, often without intuitive explanation or guiding diagrams.**
> > **The writing, particularly in the methodology section, reads like a technical report rather than a scientific paper. Many sentences could be made clearer by explaining the motivation and intuition behind each modeling choice.**
>
> Thank you for the valuable comment.
> * **We have modified the presentation of Section 3.3 and 3.4** following your suggestions of **adding intuitive motivation and intuition behind each modeling choice**. Specifically, L96-L97 introduces the overall motivation, and L263, L269-L271, L289-L290, L314-L315, and L324-l325 briefly explain the motivation behind each key design. Some examples are shown below:
>
> > L96-L97: LPR is motivated by encoding the optical priors embedded in PSFs, enabling their direct retrieval from degraded images to guide aberration correction.
> > L263: To represent PSFs as an information modality that is pixel‑aligned with the degraded image $I_{OD}\in\mathbb{R}^{H{\times}W{\times}3}$, PSF kernels are ...
> > L269-L271: Then, the key lies in learning optical priors from the PSF map. VQVAE learns a discrete latent prior into a codebook, providing transferable representations that can be directly retrieved. Motivated by this property, we propose PSF-VQVAE as the basis for LPR.
> > L289-L290: Given that the PSF‑VQVAE learns raw PSF features without modeling optical priors, we introduce ODN to further regularize LPR learning.
> > L314-L315: To predict latent PSF features characterizing optical priors from degraded inputs $I_{OD}$, we employ an encoder $E_{OD}$ identical to $E_{CAC}$ constrained by the learned LPR.
> > L324-L325: The latent space of FoundCAC is conditioned on the predicted PSF features for leveraging the learned optical priors.
>
> * In addition, **we have relocated "Related" Work section from the Appendix to the main body.** The subsection “Representation of degradation priors” (L144-L161) reviews prior studies and explains how they informed our design. We believe the earlier version lacked sufficient discussion of related work, which made our design rationale harder to follow.
> * We rename Section 3.3 and 3.4 as "Stage I: ..." and "Stage II: ..." to sharpen the logical flow of the method.
> * Our paper also included **a specific Section 3.1 (L181-L195) to illustrate the design motivation**.
>
> Building on the original paper’s clear data flow formulations, we hope that the revised manuscript is logically coherent and easy to follow.

---

> ### Author Response · Authors · 2025-11-20
> **Official Response by Authors -- Part 2**
>
> > **Figures 1 and 2 are overly complex and fail to provide a clear overview of the system. They require significant prior knowledge of the text to decode. They are repeatedly cited as explanatory ("as shown in Figure 1 (a)...", "as shown in Figure 2 (a)...") when they in fact demand the textual explanation to be understood. Key design flows and data construction steps should be broken into simpler, sequential diagrams.**
>
> Thank you for your suggestions. **We have modified the mentioned Figures, please see Figure 1, 2, and 3 in the revised paper.**
>
> * The Figure 1 has been broken into simpler and sequential diagrams (Figure 1 and Figure 2 in the revised paper). Concretely, the revised Figure 1 omits the complex data construction workflow and foregrounds the challenges the overall framework addresses. We also add Figure 2 to detail our data contributions, including the expansion of design specifications and the hybrid sampling basis derived from lens imaging quality.
> * For the original Figure 2 (now Figure 3), we have removed abbreviations to improve readability. We retain both stages in one figure because the Stage I representation is directly consumed in Stage II. With explicit data flow and symbols for key intermediate features, the figure, combined with the formal equations in the text, should make the pipeline clear to readers.
>
> > **The baseline OmniLens method is introduced too late in the text. Readers unfamiliar with the earlier work will struggle to contextualize what is actually new.**
>
> **We agree that** the OmniLens baseline should be better introduced, and **have made the following revisions** which can be found in the revised paper:
> * In L64-66, L68-69, and L72-73, we introduce the core ideas and key techniques of OmniLens and delineate the specific shortcomings of prior frameworks that our approach addresses.
> * As noted above, the revised Figure 1 succinctly highlights the data and model challenges in the OmniLens framework that OmniLens++ resolves, serving as a clear comparison between OmniLens++ and OmniLens.
> * The supplemented Section 2 on Related Work offers a focused survey of lens aberration correction (L122–L143), outlining the strengths and limitations of the OmniLens baseline and other LensLib‑PT approaches, and clarifying the motivation behind OmniLens++. This added background should help readers transition smoothly into our idea.
>
>
> > **Clarifying that AODLibpro is newly proposed and constructed in this work would strengthen the narrative.**
>
> We appreciate the suggestion and concur that the expression of AODLibpro should be strengthened, given that **it contains a novel data construction pipeline**. First, we clarify that our data construction borrows only the idea from OmniLens of using advanced automatic design to batch generate high quality lens samples. **The concrete pipeline that follows, in particular the core lens sampling strategy, is entirely new.** Our image quality based optical degradation quantification and hybrid sampling basis are the keys that give AODLibpro its scalability. **The specific manuscript updates are as follows**:
> * In L79-L84, we have revised the AODLibpro exposition to foreground its core idea of quantifying optical degradation severity and spatial variation trends from an image quality perspective, underscoring its novelty.
> * The updated Figures 1 and 2 provide intuitive views of AODLibpro’s advantages and core technical pipeline.
> * In Section 3.2 (L221–L223, L238–L239), we also added specific motivation behind AODLibpro to further highlight its originality.

---

> ### Author Response · Authors · 2025-11-20
> **Official Response by Authors -- Part 3**
>
> We have carefully responded to each of your concerns raised in the "Questions" and hope to address your doubts about our paper.
>
>
> > **Q1: The Optical Image Quality (OIQ) metric is central to AODLibpro and the evaluation, but its motivation and validation are unclear. Could the authors justify why OIQ is needed for assessing blind lens aberration correction and show that it correlates better with perceptual or optical quality than PSNR or SSIM?**
>
> Thank you for the insightful comments. We hope the following justifications will help reinforce your recognition of the design of OIQ.
>
> * **Motivation for OIQ.**
>     1. **We first clarify that OIQ is not a metric to evaluate aberration correction performance but a quantification protocal for the degradation.** It is motivated by the limitation that lens design indicators (for example RMS) cannot characterize lens specific degradation patterns. OIQ instead evaluates image quality across sub FoVs to quantify severity and spatial variation trends.
>     2. We strengthened this motivation in L79–L84 and L221–L223 as mentioned in the previous response.
>     3. We also detailed the design rationale for sampling basis in Section 3.1 (L170–L180).
>
> * **Validation of OIQ.**
>     1. **Section 4.4 (L480–L502), Table 6, and Figure 5 provided clear justification for why OIQ is needed.** Replacing the OIQ based hybrid basis with simple RMS based basis yields a LensLib that undercovers possible degradation patterns, producing models that generalize poorly to real-world lenses with limited scalability.
>     2. We have strengthened the motivation for OIQ by emphasizing an image quality based view to quantify degradation (L221-L229). Our analysis therefore examines whether this principle remedies the undercoverage and weak scalability of RMS based sampling.
>     3. The implementation in this paper is a prototype of OIQ. We adopt intuitive evaluation protocals that capture fidelity and optical characteristics, including PSNR and SSIM you mentioned.
>
> > **Q2: The experiments seem simulation-heavy. Are there quantitative evaluations on real aberrated captures, not just qualitative examples?**
>
> Thank you for the constructive suggestion. **We have added results of non-reference metrics CLIPIQA, NIQE, and MANIQA on RealLens-Snap** in Appendix G.5 (L1423 to L1450) and Table 12. The results verify that OmniLens++ maintains the strongest overall generalization, which is consistent with Figure 4. For completeness, we also include the details below (the table is extensive, we recommend reading it in the manuscript).
> ***
> **Table 12: Per‑lens numerical evaluation for competing blind lens aberration correction methods on RealLens-Snap.**
>
> | Method | Single-Lens-I  CLIPIQA↑ | Single-Lens-I  NIQE↓ | Single-Lens-I  MANIQA↑ | Single-Lens-II  CLIPIQA↑ | Single-Lens-II  NIQE↓ | Single-Lens-II  MANIQA↑ | CAYA 50mm f1/4  CLIPIQA↑ | CAYA 50mm f1/4  NIQE↓ | CAYA 50mm f1/4  MANIQA↑ |
> |:--|:--:|:--:|:--:|:--:|:--:|:--:|:--:|:--:|:--:|
> | Fast two-step | 0.341 | 3.903 | 0.218 | 0.341 | 4.907 | 0.227 | 0.324 | 3.835 | 0.180 |
> | Universal IR | 0.470 | 3.548 | 0.310 | 0.407 | 5.575 | 0.282 | 0.411 | 3.843 | 0.266 |
> | ZEBASELib-PT | 0.336 | 5.493 | 0.211 | 0.319 | 7.265 | 0.213 | 0.299 | 5.966 | 0.202 |
> | OmniLens | 0.383 | 4.789 | 0.289 | 0.392 | 5.718 | 0.327 | 0.376 | 4.311 | 0.277 |
> | **OmniLens++** | 0.398 | 4.061 | 0.272 | 0.456 | 4.792 | 0.310 | 0.393 | 3.929 | 0.273 |
>
> | Method | Nano-Optics  CLIPIQA↑ | Nano-Optics  NIQE↓ | Nano-Optics  MANIQA↑ | Canon 24mm f1/4  CLIPIQA↑ | Canon 24mm f1/4  NIQE↓ | Canon 24mm f1/4  MANIQA↑ | Sony 18135  CLIPIQA↑ | Sony 18135  NIQE↓ | Sony 18135  MANIQA↑ |
> |:--|:--:|:--:|:--:|:--:|:--:|:--:|:--:|:--:|:--:|
> | Fast two-step | 0.342 | 7.013 | 0.231 | 0.469 | 3.292 | 0.343 | 0.560 | 2.826 | 0.303 |
> | Universal IR | 0.389 | 8.172 | 0.263 | 0.433 | 3.388 | 0.332 | 0.489 | 3.050 | 0.317 |
> | ZEBASELib-PT | 0.360 | 9.522 | 0.299 | 0.519 | 4.182 | 0.359 | 0.512 | 4.264 | 0.322 |
> | OmniLens | 0.332 | 8.735 | 0.285 | 0.492 | 3.971 | 0.369 | 0.499 | 3.955 | 0.328 |
> | **OmniLens++** | 0.370 | 8.126 | 0.266 | 0.538 | 3.790 | 0.339 | 0.500 | 3.395 | 0.315 |
>
> > To more fairly and comprehensively evaluate the performance of representative blind aberration correction pipelines on real-snapped data RealLens-Snap, we conduct quantitative evaluation using common non-reference image quality metrics CLIPIQA, NIQE, and MANIQA, with results shown in Table 12. Consistent with the results in Figure 4, even though these metrics exhibit some instability, the proposed OmniLens++ still performs better overall, highlighting its stronger generalization.
> ***

---

> ### Author Response · Authors · 2025-11-24
> **Follow up**
>
> Dear Reviewer Q7Qo:
>
> Thank the valuable comments on our paper, which provided insights that can help us revise our work.
>
> We have provided a response and a revised paper, hope they could address your concerns. Also, we would like to know whether our revised statements and improved figures help you clearly understand our work. Your invaluable feedback and suggestions are greatly welcomed to help us better refine our work.
>
> Thank you again for your devotion to the review. If all the concerns have been successfully addressed, please consider raising the scores after this discussion phase.
>
> Best,
>
> Submission#3893 Authors

---

> ### Author Response · Authors · 2025-11-27
> **Looking forward to your reply**
>
> Dear reviewer Q7Qo:
>
> Since the rebuttal period has drawn to a close, would you please check out our comments and see if your questions have been successfully addressed? If so, please consider increasing the score. Or if there are more questions and concerns, please directly raise them, we will try our best to solve them.
>
> Considering that your suggestions and concerns are most at the presentation level, we suggest that you directly read the final revised version of our paper to review our modifications. You can download the updated PDF by clicking the button next to the paper title at the top of the page to check our modification following the marked points in our responses.
>
> Best,
>
> Submission#3893 Authors

---

> > ### Comment · Reviewer_Q7Qo · 2025-11-27
> > **Follow-up review**
> >
> > Thanks for the detailed rebuttal and the revision effort. A few of the presentation/evidence issues I raised still seem unresolved in the current manuscript.
> >
> > > We appreciate the suggestion and have removed the unnecessary abbreviations OD and CAC in the latest revision to improve readability.
> >
> > In the current draft, OD and CAC are still actively used and even re-introduced as abbreviations. For example, OD still appears (e.g., around line 172, 232-237, 294, 315; Fig. 3; Eq. 4-5), and CAC still appears (e.g., Fig. 3; around line 311, 315, 324-340), and they remain listed in the abbreviation table.
> > So right now it reads less like "removal" and more like "the terms persist, but sometimes without consistent (re-)definition near first use".
> >
> > > We retain ODN and OIQ because they denote our proposed module and method, and their abbreviations facilitate concise notation and formula definitions.
> >
> > For ODN, my concern isn't the abbreviation itself but the semantic framing. As currently written, "Optical Degradation Network" can imply the degradation is "caused by the network", when the interesting technical point is that you learn a degradation-conditioned latent/fusion mechanism and use it as a constraint/regularizer. Your method section actually describes ODN as modeling the clear to OD mapping conditioned on quantized PSF features via a fusion module.
> > That's totally fine technically, but the focus should be on what's novel: the PSF-conditioned fusion and how it shapes the codebook/latent space.
> > Concretely, the "fusion module" explanation (right now buried in the ODN definition and later appendix references) should be moved into the main ODN paragraph, and I'd encourage naming submodules by function (image encoder/decoder, degradation module, PSF-conditioned fusion) to avoid the impression of "the network is the optics".
> >
> > > Table 9 in Appendix.B lists the full expansions of all abbreviations used in the paper to facilitate smoother reading.
> >
> > A glossary is helpful, but it doesn't solve the reading-flow problem. The main fix is still to reduce jargon density and only abbreviate what is repeatedly used in equations (and then use it consistently).
> >
> > > We have modified the mentioned Figures, please see Figure 1, 2, and 3 in the revised paper.
> >
> > Figures 1-2 may be improved, but Figure 3 (former Figure 2) hasn't changed compared to the original manuscript and is still visually overloaded and it relies heavily on the reader already knowing the pipeline.
> >
> > > We have added results of non-reference metrics CLIPIQA, NIQE, and MANIQA on RealLens-Snap in Appendix G.5 (L1423 to L1450) and Table 12. The results verify that OmniLens++ maintains the strongest overall generalization, which is consistent with Figure 4.
> >
> > Adding real-data quantitative evaluation is absolutely the right move. However, based on the table summary you provided, OmniLens++ is best in 3/18 cases and second best in 6/18, which reads more like "competitive" than "strongest overall". Also: highlighting best/second-best in the table improves readability.
> >
> > Overall: the revision is moving, but I'm not yet convinced the manuscript is genuinely easier to read (abbreviations still pervasive, now less defined), and the new real-data quantitative results currently look mixed rather than clearly dominant unless you justify the aggregation and metric relevance more carefully.

---

> ### Author Response · Authors · 2025-11-29
>
> We sincerely appreciate your time and constructive feedback. Incorporating your latest suggestions, we have updated the manuscript and highlighted all new revisions in red. We believe that this final version has addressed all your concerns. The main updates are summarized below:
> > **In the current draft, OD and CAC are still actively used and even re-introduced as abbreviations. For example, OD still appears (e.g., around line 172, 232-237, 294, 315; Fig. 3; Eq. 4-5), and CAC still appears (e.g., Fig. 3; around line 311, 315, 324-340), and they remain listed in the abbreviation table. So right now it reads less like "removal" and more like "the terms persist, but sometimes without consistent (re-)definition near first use".**
>
> We thoroughly audited the manuscript and removed OD and CAC abbreviations throughout the text, equations, and notation. We also renamed "OD-Class" to "Spatial-Class" to replace opaque abbreviations with clearer terms. The only remaining use of CAC is in the model name FoundCAC.
>
> > **For ODN, my concern isn't the abbreviation itself but the semantic framing. As currently written, "Optical Degradation Network" can imply the degradation is "caused by the network", when the interesting technical point is that you learn a degradation-conditioned latent/fusion mechanism and use it as a constraint/regularizer. Your method section actually describes ODN as modeling the clear to OD mapping conditioned on quantized PSF features via a fusion module. That's totally fine technically, but the focus should be on what's novel: the PSF-conditioned fusion and how it shapes the codebook/latent space. Concretely, the "fusion module" explanation (right now buried in the ODN definition and later appendix references) should be moved into the main ODN paragraph, and I'd encourage naming submodules by function (image encoder/decoder, degradation module, PSF-conditioned fusion) to avoid the impression of "the network is the optics".**
>
> Following your suggestions, we have renamed the "ODN" constraint to “PSF‑conditioned regularizer” and reorganized the expression of its core idea (L280-L289). The module where PSF features interact with the network has been renamed “PSF‑conditioned fusion” and we present its specific architecture and design motivation directly in the main text (L287-L289, Figure 3).
>
> >**A glossary is helpful, but it doesn't solve the reading-flow problem. The main fix is still to reduce jargon density and only abbreviate what is repeatedly used in equations (and then use it consistently)**.
>
> We have also removed the OD and CAC abbreviations from the abbreviation table, and ensured that neither the main text nor the appendix uses these notations. Instead, we adopt more rigorous definitions and phrasing to improve readability.
>
> > **Figures 1-2 may be improved, but Figure 3 (former Figure 2) hasn't changed compared to the original manuscript and is still visually overloaded and it relies heavily on the reader already knowing the pipeline.**
>
> We have split the original Figure 2 into Figure 3 and Figure 4 to present the two training stages clearly and succinctly. Both figures were redesigned to clarify the data flow, link each core objective to its explicit equation, and use simplified module icons.  In addition, the specific structure of the PSF-conditioned fusion and the newly defined notation you are concerned about are shown in the figures. We believe these changes will help readers better understand the overall training data flow and align it with the equations in the text for an intuitive understanding of the procedure.
>
> > **Adding real-data quantitative evaluation is absolutely the right move. However, based on the table summary you provided, OmniLens++ is best in 3/18 cases and second best in 6/18, which reads more like "competitive" than "strongest overall". Also: highlighting best/second-best in the table improves readability.**
>
> We marked the top‑3 results in Table 12 and added more detailed analyses and explanations of the non‑reference metric evaluations (L1428-L1436) to ensure the rigor of the claims. The specific content is as follows:
>
> > Consistent with the results in Figure 5, even though these metrics exhibit some instability, the proposed OmniLens++ still performs better overall, highlighting its competitive generalization. Specifically, most of OmniLens++ results rank in the top 3 across these metrics (16/18), outperforming the second best method, Universal IR (14/18). Additionally, OmniLens++ shows no outliers in its results, that is, no cases with particularly poor performance. Since Universal IR uses S3Diff, a diffusion based generative model whose outputs exhibit unrealistic sharpening yet enjoy inherent advantages on these non-reference metrics (Zhang et al., 2024), these results indicate the favorable robustness and generalization of OmniLens++.

---

### Official Review · Reviewer_QyYn · 2025-10-31

**Soundness:** 3
**Presentation:** 4
**Contribution:** 3
**Rating:** 6
**Confidence:** 3

**Summary:**

This paper introduces OmniLens++, a new framework for blind lens aberration correction that leverages a large-scale LensLib pre-training pipeline and a Latent PSF Representation (LPR) module. This paper aims to overcome two limitations in existing approaches: insufficient scalability of optical degradation data and the lack of explicit degradation priors in blind correction models. To address these, the authors construct AODLibpro, a uniformly sampled lens library with enriched optical specifications, and design an LPR-guided CAC (Computational Aberration Correction) model using a VQVAE-based latent PSF codebook and an Optical Degradation Network for prior regularization. Extensive experiments on synthetic and real lenses demonstrate state-of-the-art performance and strong generalization to unseen aberrations. While the technical pipeline is well-executed and the experimental evaluation is compelling, the paper would benefit from a more
thorough explanation of the motivation behind the latent PSF representation and the specific advantages of the VQVAEODN combination.

**Strengths:**

The paper presents a technically solid and conceptually novel contribution by linking large-scale lens data construction with degradation-prior learning in a blind setting. The experimental evaluation is comprehensive and compelling, demonstrating state-of-the-art performance across both simulation and real-world benchmarks. The writing is clear and well-structured, making the technical contributions straightforward to follow.

**Weaknesses:**

Although the framework is comprehensive and experimentally solid, I find the overall technical novelty is relatively limited.
The proposed VQVAE-based design represents a straightforward structural adaptation rather than a fundamentally new methodological contribution. While the integration of the Optical Degradation Network (ODN) to regularize the latent space is a thoughtful addition, it largely follows standard practice in generative modeling and degradation-aware representation learning. The author should provide more detailed discussion or ablation study.

**Questions:**

1. Could the authors provide a more detailed ablation study on the hyperparameter settings? For example, on page
17, the LPR module is described as using a VQVAE with a codebook size of K=1024 and a latent feature
dimension of n_z=256. It would be helpful to understand how these choices affect model performance and whether
the results are sensitive to variations in these parameters.

2. While the paper demonstrates strong generalization across various lens types, it remains unclear how OmniLens++
performs under more extreme imaging conditions, such as low-light environments or high dynamic range scenes.

3. In line 362, the authors mention the suppression of purple fringing caused by chromatic aberration. Could they
provide a more detailed explanation of how this is achieved? This would help reviewers and readers better
understand the model’s capability in handling chromatic aberrations and its implications for real-world applications.

---

> ### Author Response · Authors · 2025-11-20
> **Official Response by Authors -- Part 1**
>
> We would like to sincerely thank the reviewer for the constructive and detailed comments on our work. We take every comment seriously and hope our response can address the your concerns. If there are any remaining questions, we are more than happy to address them.
>
> > **Overall comment: While the technical pipeline is well-executed and the experimental evaluation is compelling, the paper would benefit from a more thorough explanation of the motivation behind the latent PSF representation and the specific advantages of the VQVAEODN combination.**
>
> Thank you for your overall recognition of our work and for your valuable suggestions to further improve the paper. Regarding the highlights you mentioned, we have made corresponding revisions in the manuscript and hope this addresses your concerns.
>
> * The motivation behind the LPR
>   1. We first clarify that **Section 3.1 (L181–L195) details the motivation for LPR**. The motivation stems from experimental findings (Table 1), where PSFs provide effective guidance for aberration correction but are unavailable in the blind setting. Predicting PSF features in latent space offers a viable alternative. These observations motivate our development of LPR.
>   2. To clarify LPR’s motivation and improve the narrative flow, **we have added supporting text in Section 1 and Section 3.3.** Specifically, L96-L97 introduces the overall motivation, and L263, L269-L271, and L289-L290 briefly explain the motivation behind each key design of LPR.
>   3. **We have relocated "Related" Work section from the Appendix to the main body.** The subsection “Representation of degradation priors” (L144-L161) reviews prior studies and explains how they informed our design of LPR, which we expect will further clarify LPR’s design rationale for readers.
>
> * The specific advantages of the VQVAEODN combination.
> We clarify that evaluating PSF-VQVAE and ODN is central to our experiments. **Table 4 and L457–L465 present targeted analyses.**  Beyond PSF-VQVAE and ODN, we benchmark alternative representations including direct PSF feature prediction and using CAC instead of ODN as the auxiliary constraint. These studies substantiate the technical advantages of LPR. **We have also provided the visualization of the learned representation in Appendix.G.4 (L1391-L1405) and Figure 19.** **The suggested hyperparameter explorations have been added to Appendix.G.3 (L1373–L1390)**, which will be discussed below.
>
> > **Comment on weaknesses: I find the overall technical novelty is relatively limited. The proposed VQVAE-based design represents a straightforward structural adaptation rather than a fundamentally new methodological contribution. While the integration of the Optical Degradation Network (ODN) to regularize the latent space is a thoughtful addition, it largely follows standard practice in generative modeling and degradation-aware representation learning. The author should provide more detailed discussion or ablation study.**
>
> We appreciate your thoughtful insights.
> * **We have added detailed explanations of the motivation and rationale for PSF-VQVAE and ODN in the previous response** to better convey the technical novelty.
> * Beyond motivation, **both components are supported by rigorous experimental validation (Table 4 and L457–L465, Figure 19 and L1375–L1390)**. As requested, we also include additional hyperparameter ablations, which are reported in the subsequent responses.
> * We further report experiments evaluating **the interaction between the data design and the model design (Table 8, L517–L523)**. The results indicate that LPR effectively exploits AODLibpro's scalability, and this synergy highlights the novelty of the overall framework.

---

> ### Author Response · Authors · 2025-11-20
> **Official Response by Authors -- Part 2**
>
> > **Q1: Could the authors provide a more detailed ablation study on the hyperparameter settings? For example, on page 17, the LPR module is described as using a VQVAE with a codebook size of K=1024 and a latent feature dimension of n_z=256. It would be helpful to understand how these choices affect model performance and whether the results are sensitive to variations in these parameters.**
>
> * Thank you for the constructive suggestions. **We have added a codebook size ablation** to investigate how to learn effective LPR. **The study appears in Appendix G.3 (Table 11, L1375 to L1390)**, and we provide the key results and analysis below.
>
> ***
> **Table 11: Ablations on codebook size in LPR**
>
> | K    | PSNR  | LPIPS  |
> |:---: |:----: |:-----: |
> | 512  | 28.53 | 0.1301 |
> | 2048 | 28.78 | 0.1297 |
> | **1024** | 28.67 | 0.1277 |
>
> > We sweep the codebook size $K$ to test whether allocating more entries for key latent PSF features benefits LPR. Table 11 shows no meaningful gains for FoundCAC from larger $K$. For the current PSF diversity in AODLibpro, $K=1024$ appears sufficient to capture latent optical priors, whereas larger $K$ likely accumulates ineffective codes, complicates matching, and yields less reliable PSF guidance. A promising direction is to devise training strategies for larger codebooks that learn more effective degradation representations.
> ***
> * We set the latent dimension to 256, consistent with common VQ-based image restoration settings[1-3] and to enable convenient reuse of pretrained codebooks. This also matches the channel width of the model’s deepest features. Because altering this dimension means the changes of the baseline, we believe it would not affect the current experimental conclusions.
>
> [1] Chen, Chaofeng, et al. "Real-world blind super-resolution via feature matching with implicit high-resolution priors." Proceedings of the 30th ACM International Conference on Multimedia. 2022.
> [2] Wu, Rui-Qi, et al. "Ridcp: Revitalizing real image dehazing via high-quality codebook priors." Proceedings of the IEEE/CVF conference on computer vision and pattern recognition. 2023.
> [3] Chen, Shiqi, et al. "Mobile image restoration via prior quantization." Pattern Recognition Letters 174 (2023): 64-70.

---

> ### Author Response · Authors · 2025-11-20
> **Official Response by Authors -- Part 3**
>
> > **Q2: While the paper demonstrates strong generalization across various lens types, it remains unclear how OmniLens++ performs under more extreme imaging conditions, such as low-light environments or high dynamic range scenes.**
>
> Thank you for the thoughtful suggestion. **We agree that** testing blind aberration correction under extreme conditions such as low light, high dynamic range, and noise is an important direction for future work, and **we have added a discussion in Section 5 (L533–L539).**
> **Additional clarification:**
> * Our work targets at building a universal model that robustly handles PSF induced lens degradations with diverse spatial variation patterns and severities, which is both academically and practically valuable. Hence the evaluation emphasizing coverage across optical degradation patterns is sufficient to support this research.
> * Some challenging real-world scenes are included in the evaluation. For example, in Figure 4, the results of a large aperture lens under outdoor stray light, and a metalens under low light condition are provided. These results show that our method corrects PSF blur even when other degradations are present, although it does not directly remove those additional factors.
>
> We hope these clarifications better convey the motivation and positioning of our work.
>
> > **Q3: In line 362, the authors mention the suppression of purple fringing caused by chromatic aberration. Could they provide a more detailed explanation of how this is achieved? This would help reviewers and readers better understand the model’s capability in handling chromatic aberrations and its implications for real-world applications.**
>
> Thank you for your suggestion. **We have supplemented an analysis at the end of Appendix D.2** clarifying why OmniLens++ effectively addresses chromatic aberration **(L953–L959, Figure 10)**. The key is that AODLibpro naturally yields many lens samples with pronounced chromatic aberration, which equips the trained model with effective chromatic aberration correction ability. The complete analysis is shown below.
>
> > Then, because the configuration of EAOD algorithm  does not include settings for a cemented doublet structure, the optimization imposes no strong constraint on chromatic aberration, which causes many generated samples to exhibit noticeable chromatic aberration. Figure 10 shows the distribution of chromatic aberration severity in our AODLibpro Train, from which it can be seen that a considerable portion of the training samples reveal obvious chromatic aberration. We believe that these chromatically degraded samples in the training set endow the model with the ability to handle chromatic aberration, enabling it to effectively address purple fringing across the test cases.

---

> ### Author Response · Authors · 2025-11-24
> **Follow up**
>
> Dear Reviewer QyYn:
>
> Thank the valuable comments on our paper, which provided insights that can help us revise our work.
>
> We have provided a response and a revised paper, hope they could address your concerns. Also, we would like to know if there are more concerns about the content of the paper. Your invaluable feedback and suggestions are greatly welcomed to help us better refine our work.
>
> Thank you again for your devotion to the review. If all the concerns have been successfully addressed, please consider raising the scores after this discussion phase.
>
> Best,
>
> Submission#3893 Authors

---

> > ### Comment · Reviewer_QyYn · 2025-11-27
> >
> > The authors' response addresses my main concerns. I keep my positive recommend. If accepted, authors must improve the final version according to the above comments.

---

> ### Author Response · Authors · 2025-11-27
>
> Dear Reviewer QyYn:
>
> Thank you very much for your thoughtful feedback! We’re delighted to hear that our reply addressed your concerns. All revisions and added experiments described in our responses have been fully incorporated into the final manuscript. **You can download the updated PDF by clicking the button next to the paper title at the top of the page to check our modification following the marked points in our responses.** We hope that after reading our final version you will further support our work. If this revised version meets your requirements, please consider raising the scores.
>
> Best,
>
> Submission#3893 Authors

---

### Author Response · Authors · 2025-11-20
**General Response**

We thank all reviewers for their detailed reviews and suggestions!

We have updated the manuscript with the following revisions based on the reviewers' suggestions. All revisions in the updated version **are highlighted in blue**：
1. **Modification of main figures and presentations on motivation.**
    * The Figure 1 has been broken into simpler and sequential diagrams (Figure 1 and Figure 2 in the revised paper). Figure 2 is added to detail our data contributions.
    * The introduction of OmniLens baseline is enhanced (L64-66, L68-69, L72-73) with a clear comparison between Omnilens++ and OmniLens (Figure 1).
    * The motivation behind the data construction (L79-L84, L221–L223, L238–L239) and model design (L96-L97, L263, L269-L271, L289-L290, L314-L315, L324-L325) is better expressed.
    * Section 2 Related Work (L120-L161) is added to survey prior work and foreground the core ideas and innovations of our method.
    * Unnecessary abbreviations are removed.
2. **Additional experiments and evaluation.**
    * Ablations on codebook size (L1373-L1388, Table 11).
    * Quantitative evaluation of real-world results (L1423-L1450, Table 12).

3. **Additional discussion.**
    * Discussion on chromatic aberration (L953-L959, Figure 10).
    * Discussion on future work of aberration correction under other optics-coupled degradations, such as depth of field, low light, and under display imaging (L535-L538).
    * Discussion on future work of fine-tuning pipeline (L538-L539).

---

### Meta-Review · Area_Chair_Bp5f · 2026-01-06

**Summary:**

This manuscript proposes a framework for blind aberration correction, OmniLens++. It is build on Lens Library Pre-Training (lensLib-PT). Thereby the paper introduces a library (AODLibpro) that expands the design space with aspheric surfaces and image-plane perturbations to increase corruption diversity.  Furthermore, it introduces a latent PSF Representation (LPR) to include PSF knowledge into a blind correction pipeline. A foundational aberration-correction model (FoundCAC) is trained based on LPR. The model is evaluated on  RealLens-Sim and real photographs and yields strong results.
Minor comment by the AC: an evaluation on data from Mueller et al., TPAMI 2025 might also make sense: https://ieeexplore.ieee.org/document/11205303

**Reviewer Concerns:**

overall technical novelty is relatively limited (Reviewer QyYn, Reviewer 6hkZ).
more detailed discussion and ablation needed (Reviewer QyYn). --> addressed
presentation is needlessly hard to parse (Reviewer Q7Qo) --> partly addressed
presentation needs to be improved (Reviewer 6hkZ)
stray light simulation should be included (Reviewer 6hkZ) --> not addressed

The critical concerns are at least partly addressed and in particular when the presentation is concerned, the authors clearly improved the manuscript to a large extent, making it a borderline case.
Questions regarding technical details have been addressed by the authors.

**Reviewer Scores:**

This is a borderline case the reviewers gave borderline ratings before the discussion and were carefully positive after the discussion. In fact, several of the suggested improvements have been implemented but not all. Even if all recommendations regarding the writing and structure of the documents had been followed, I would expect this paper to remain a borderline case with limited technical contribution and still somewhat hard to parse. I therefore recommend to reject to reject the paper and hope the authors can further improve the manuscript organization. The suggested experiments on stray light simulation would be nice but the AC agrees that this would significantly expend the scope of the paper.

---

### Decision · Program_Chairs · 2026-01-26

Reject